# Digital quantum simulation of bosonic systems and quantum complementarity

Victor P. Brasil [ID],[1, *] Diego S. Starke [ID],[1, †] and Jonas Maziero [ID][1, ‡]

[1]*Physics Department, Federal University of Santa Maria, 97105-900, Santa Maria, RS, Brazil*
(Dated: September 30, 2025)

Digital quantum simulation (DQS) has emerged as a powerful approach to investigate complex quantum systems using digital quantum computers. Such systems, like many-particle bosonic systems and intricate optical experimental setups, pose significant challenges for classical simulation methods. In this paper, we utilize a general formalism for the DQS of bosonic systems, which consists of mapping bosonic operators to Pauli operators using the Gray code, in order to simulate interferometric variants of Afshar's experiment—an intricate optical experiment—on IBM's quantum computers. We investigated experiments analogous to Afshar's double-slit experiment performed by Unruh and Pessoa Júnior, exploring discussions on the apparent violation of Bohr's complementarity principle when considering the entire experimental setup. Based on the aforementioned experiments, we construct a variation of a delayed-choice setup. We also explore another experiment starting with a two-photon initial state. Finally, we analyze these experiments within the framework of an updated quantum complementarity principle, which applies to specific quantum state preparations and remains consistent with the foundational principles of Quantum Mechanics.

Keywords: Digital quantum simulation, Afshar's experiment, Unruh's experiment, Bohr's complementarity principle, Quantum complementarity relations

## I. INTRODUCTION

Feynman [1] proposed that quantum systems could be used to simulate other quantum systems. This proposal is capable of addressing the limitations inherent in classical computational methods for quantum system simulation, especially concerning the exponential increase in parameters necessary for describing such systems. Today, it is understood that quantum systems can be simulated using quantum devices designed for specific problem-driven evolutions [2]. Lloyd [3] expanded on this idea by establishing the theoretical foundation of quantum computers as universal machines capable of addressing a broad range of problems. These devices leverage quantum phenomena such as superposition and entanglement to solve both classical and quantum problems more efficiently than classical computers [4, 5]. However, realizing this potential requires effective methods for mapping problems onto quantum computers, a task that is generally difficult.

A significant challenge in the mapping process lies in the digital quantum simulation (DQS) of bosonic systems [6–8]. These systems are characterized by an infinite-dimensional Hilbert space, necessitating a two-step simulation approach [6]: *i*) encoding bosonic states into qubit states, and *ii*) establishing an isomorphic mapping of bosonic field operators to linear operators within the qubit state space. Following this framework, the authors of Refs. [9–11] adopted a methodology that utilizes the Gray codification to map bosonic operators to Pauli operators. To demonstrate the effectiveness of this formalism, Mohan *et al.* [9] constructed a quantum circuit for simulating a beam splitter and implemented the Hong-Ou-Mandel (HOM) interference experiment.

The investigation of bosonic systems holds significant importance across numerous research domains, including biology [12–15], chemistry on bosonic devices (a promising model of quantum computation) [16–19], condensed matter physics [20–24] and quantum optics [25–28]. Our study aims to simulate bosonic systems as a means to explore the foundations of QM; in particular, we are interested in DQSs for replicating the experimental setups analogous to the modified double-slit experiment conducted by Afshar *et al.* [29] performed with Mach-Zehnder interferometer. Afshar's experiment suggests that it is possible to observe both aspects of wave-particle duality in the same experimental setup, thereby asserting a breach of Bohr's complementarity principle. Building upon this idea, a modified version of Unruh's experiment [30] is considered, followed by an extended experimental configuration, Pessoa Júnior's experiment [31].

Our main goal is therefore to explore the formalism for simulating bosonic systems and its applicability to advanced experimental configurations, such as that of Ref. [31]. In particular, while the modified Unruh's experiment and the alternative version of Pessoa Júnior's setup can be simulated without second quantization if blockers that appear in

———

*Electronic address: victor.brasil@acad.ufsm.br
†Electronic address: starkediego@gmail.com
‡Electronic address: jonas.maziero@ufsm.br

these investigations are disregarded, the inclusion of blockers requires a more elaborate treatment. This motivates the use of a formalism capable of consistently capturing both the presence and absence of absorption in the blockers within a single quantum circuit. Additionally, our analysis extends the modified Unruh's experiment to the two-photon case, providing a richer scenario for assessing the effectiveness of this approach.

We also analyze the experiments in the light of an updated quantum complementarity principle (QCP), synthesized in Ref. [32]. Given that the complementarity relations (CRs) derived for wave-particle duality are done so through the postulates of QM, a violation of this type of CR as stated by Afshar would imply some inconsistency in the foundations of quantum theory. Furthermore, we will argue that if one considers Pessoa Júnior's procedure, it is possible to construct a delayed-choice experiment that allows the dual behavior of a quanton to be modified even after it has left one of the Mach-Zehnder interferometers.

The organization of this paper is as follows. In Section II A, the formalism used in Ref. [9] is revisited. Section II B reviews the quantum complementarity principle (QCP), Afshar's experiment, and the arguments used by that author to claim the violation of Bohr's complementarity principle. Sections II C and II D explore different versions of nested MZI as described by Pessoa Júnior in Ref. [31] as an analogous version of Afshar's experiment. Sections II C 1 and II D 1 detail the DQS of these experiments, respectively. In Section II E we provide a new analysis of the modified Unruh's experiment by considering a two-photon initial state. In Section II F, we expand on the discussion of delayed-choice experiments and build a new kind of the delayed-choice experiment that arises in the formulation constructed by Pessoa Júnior. Following, the results are analyzed employing the QCP framework. Final remarks are presented in Sec. III. The demonstrative results compiled in this study to develop the histograms and the plots are available in Ref. [63].

## II. DIGITAL QUANTUM SIMULATION (DQS) OF INTERFEROMETRIC VERSIONS OF AFSHAR'S EXPERIMENT AND THE QUANTUM COMPLEMENTARITY PRINCIPLE

### A. Formalism for the DQS of bosonic systems

To illustrate the formalism for constructing digital quantum simulations of bosonic systems using the Gray code [9–11], let us first consider a symmetric lossless beam splitter (BS) with input modes A and B, output modes C and D, and with the property that reflected beams pick up a phase of $\pi/2$ rad (Fig. 1). The unitary transformation for this type of BS is represented by [33]

$$U_{\mathrm{BS}} = \exp\{i\theta(b^\dagger a + ba^\dagger)\}, \tag{1}$$

where bosonic annihilation (creation) operators are denoted by lowercase letters $a$, $b$ ($a^\dagger$, $b^\dagger$) corresponding to the respective uppercase letter modes, and $\theta = \arctan(R/T)$ specifies the ratio between the reflection $R$ and transmission $T$ coefficients. In subsequent analysis of specific cases where $T \neq R$, we denote this optical element by BBS (biased beam splitter). The relations between input-output operators are expressed by [33]

$$
\begin{aligned}
c &= Ta + iRb, & d &= iRa + Tb, \\
c^\dagger &= Ta^\dagger - iRb^\dagger, & d^\dagger &= -iRa^\dagger + Tb^\dagger,
\end{aligned}
\tag{2}
$$

and conversely

$$
\begin{aligned}
a &= Tc - iRd, & b &= -iRc + Td, \\
a^\dagger &= Tc^\dagger + iRd^\dagger, & b^\dagger &= iRc^\dagger + Td^\dagger.
\end{aligned}
\tag{3}
$$

These relations are derived in Appendix A.

In Quantum Mechanics (QM) second quantization, the states of quantum systems (quantons) are usually described by state vectors in a Fock space $\mathscr{F}$, a natural framework for dealing with systems of indistinguishable particles [34]. In general, a Fock state of a system with $N$ particles can be represented by

$$|n_1, n_2, \cdots, n_m, \cdots\rangle \in \mathscr{F}_N, \tag{4}$$

where the first mode is occupied by $n_1$ photons, the second mode by $n_2$ photons, the $m$-th mode by $n_m$ photons and so on (with the constraint that $\sum_j n_j = N$), and $\mathscr{F}_N$ is the $N$-particle subspace of $\mathscr{F}$. The general state vector of this system is a linear superposition of these states:

$$|\psi\rangle = \sum_{n_1}\sum_{n_2}\cdots\sum_{n_m}\cdots C_{n_1,n_2,\cdots,n_m,\cdots}|n_1, n_2, \cdots, n_m, \cdots\rangle, \quad C_{n_1,n_2,\cdots,n_m,\cdots} \in \mathbb{C}. \tag{5}$$

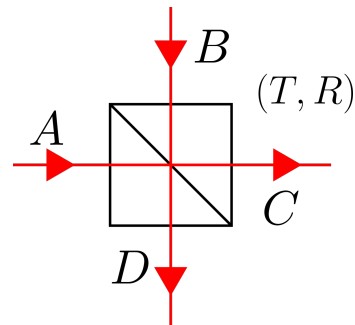

Figure 1: Beam splitter with input modes A and B and output modes C and D, characterized by transmission and reflection coefficients $T$ and $R$, respectively. Each reflection adds a phase of $\pi/2$ rad to the reflected beam.

The authors in Refs. [9–11] begin by setting the number $N$ of bosons to be simulated and then encoding each $N + 1$ Fock state into qubit states using the Gray code. Some useful properties of this representation are that adjacent numbers have a Hamming distance (number of positions at which the corresponding bits are different) equal to one and that the Gray code representation of a number can be easily obtained from its binary representation while maintaining the same number of digits. Consider an arbitrary number $m$ represented in binary, $B(m) = b_0 b_1 \cdots b_{l-2} b_{l-1}$, $b_j \in \{0, 1\}$ for $j = 0, \cdots, l - 1$, where $l$ is the length of the bit strings) and its corresponding Gray representation, $G(m)$. The most significant digit in the two representations is the same, i.e., $G_0(m) = B_0(m)$, and the subsequent digits of the Gray representation are obtained in descending order of significance by the recurrence formula

$$G_k(m) = B_k(m) \oplus B_{k-1}(m), \ k = 1, \cdots, l - 1, \tag{6}$$

where the symbol $\oplus$ denotes the "exclusive or" operation. To construct the single-mode bosonic creation operator, now consider two sequential single-mode Fock states mapped into qubit states using the Gray code,

$$|n\rangle \ \mapsto \ \big|g_0 \ \cdots \ g_{j-1} \ g_j \ g_{j+1} \ \cdots \ g_{N_q-1}\big\rangle, \tag{7}$$

$$|n+1\rangle \ \mapsto \ \big|g_0 \ \cdots \ g_{j-1} \ g_j' \ g_{j+1} \ \cdots \ g_{N_q-1}\big\rangle, \tag{8}$$

where $n \in \mathbb{N}$, $g_k \in \{0, 1\}$ for $k = 0, \cdots, N_q - 1$, $N_q = \lceil \log_2(N + 1) \rceil$ is the number of qubits per mode needed to simulate a system with $N$ bosons and the numbers $n$ and $n+1$ differ in the Gray representation at position $j$ ($g_j \neq g_j'$). The map between operators is then constructed by analyzing in which position the Gray code representations of $n$ and $n+1$ have different bits (if $G_i(n) \oplus G_i(n+1) = 1$, then $G(n)$ and $G(n+1)$ have different bits at position $i$ for some $i \in [0, \cdots, N_q - 1]$) and then acting a spin ladder operator $\mathcal{S}$ in the qubit of the corresponding position and projectors in the remaining qubits, for $n = 0, \cdots, N - 1$:

$$a^\dagger \mapsto \sum_{n=0}^{N-1} \sqrt{n+1} \bigotimes_{j=0}^{N_q-1} \Big( \delta_{0,G_j(n) \oplus G_j(n+1)} \ \mathcal{P}_{G_j(n)} + \delta_{1,G_j(n) \oplus G_j(n+1)} \ \mathcal{S}^\dagger_{G_j(n)} \Big) =: a_P^\dagger, \tag{9}$$

where

$$\mathcal{P}_0 = \frac{1}{2}(\mathbb{I} + Z) = |0\rangle\langle 0|, \quad \mathcal{P}_1 = \frac{1}{2}(\mathbb{I} - Z) = |1\rangle\langle 1|, \tag{10}$$

$$\mathcal{S}_0 = \frac{1}{2}(X + iY) = |0\rangle\langle 1|, \quad \mathcal{S}_1 = \frac{1}{2}(X - iY) = |1\rangle\langle 0|, \tag{11}$$

and $\mathbb{I}, X, Y$ and $Z$ are the usual Pauli matrices. The single-mode bosonic annihilation operator is thus simply

$$a \mapsto \sum_{n=0}^{N-1} \sqrt{n+1} \bigotimes_{j=0}^{N_q-1} \Big( \delta_{0,G_j(n) \oplus G_j(n+1)} \ \mathcal{P}_{G_j(n)} + \delta_{1,G_j(n) \oplus G_j(n+1)} \ \mathcal{S}_{G_j(n)} \Big) =: a_P. \tag{12}$$

Since bosonic operators only act on their respective modes (operators of distinct modes commute) and considering the systems under analysis are closed (in which the total number of bosons is conserved), the case for multi-mode systems follows directly as a tensor product structure of single-mode systems:

$$a_j^\dagger \mapsto \mathbb{I}^{\otimes j} \otimes a_P^\dagger \otimes \mathbb{I}^{\otimes m-(j+1)} \tag{13}$$

$$a_j \mapsto \mathbb{I}^{\otimes j} \otimes a_P \otimes \mathbb{I}^{\otimes m-(j+1)}, \quad j \in [0, m-1], \tag{14}$$

where $a_j^\dagger$ ($a_j$) is the creation (annihilation) operator of mode $j$ and $m \geq 1$ is the number of modes of the system.

## B. Quantum complementarity and Afshar's experiment

Bohr's complementarity principle (BCP) [35] has long been a cornerstone for understanding QM. Until recently, the exact origin of BCP within QM remained unclear. However, Ref. [36] demonstrated that a quantum version of BCP can be derived directly from the foundational postulates of QM, offering a deeper theoretical basis for this principle.

The updated QCP proposed in Ref. [32] is stated as follows: *For a given quantum state preparation $\rho_t$ at a specific time instant $t$, the wave-like and particle-like manifestations of a quanton are constrained by the quantum CR*

$$\mathfrak{W}(\rho_t) + \mathfrak{P}(\rho_t) \leq \alpha(d), \tag{15}$$

*which is derived directly from the axioms of QM*, where $\mathfrak{W}$ is a quantifier of wave-like behavior, $\mathfrak{P}$ is a quantifier of particle-like behavior, and $\alpha$ is a constant that depends only on the dimension $d$ of the quanton. Deriving QCRs of this type becomes achievable after taking into account quantum coherence [37] as a good quantifier of $\mathfrak{W}$ aligned with certain restrictions on quantum states $\rho$ (density operator) provided by quantum theory. The procedure for obtaining QCRs shows that the QCP can be uniquely derived from the QM postulates. The subscript $t$ in $\rho_t$ denotes the quantum state at a precise instant of time that marks the unitary evolution of the system, implying that the quantum state plays a crucial role in analyzing wave-particle duality, as it clearly demonstrates that QCRs must be continuously evaluated as the state evolves over time. An example of QCR is [32]

$$C_{l_1}(\rho_t) + P_{l_1}(\rho_t) \leq d - 1. \tag{16}$$

Here, $\mathfrak{W}(\rho_t) = C_{l_1}(\rho_t) = \sum_{j \neq k} |\rho_{jk}|$ is known as the $l_1-$norm quantum coherence, where $\rho_{jk}$ are the elements of the density operator $\rho_t$ on a given basis and the predictability measure is given by $\mathfrak{P}(\rho_t) = P_{l_1}(\rho_t) = d - 1 - \sum_{j \neq k} \sqrt{\rho_{jj}\rho_{kk}}$. Equality is achieved when the analyzed system is pure.

The principle can be extended to evaluate quantum correlations between a bipartite quantum system by introducing the entanglement monotone $\mathfrak{E}(\rho_t)$. For a particular choice of $\mathfrak{W}(\rho_t)$, $\mathfrak{P}(\rho_t)$ in Eq. (15), which satisfy some basic properties [38, 39], it is always possible to establish a quantum complete complementarity relation (CR) given by $\mathfrak{W}(\rho_t) + \mathfrak{P}(\rho_t) + \mathfrak{E}(\rho_t) = \alpha(d)$, also known as the triality relation [40–44]. It is worth mentioning that although in this work we explore a single bosonic system, recent developments have been reported in Ref. [45] regarding the understanding of wave-particle duality applied to many-body quantum systems involving both bosons and fermions.

Before the introduction of the QCP, the quantification of BCP was obtained through a complementarity relationship that can be expressed as

$$W + P \leq \beta \tag{17}$$

where $\beta$ is a constant, $W$ is the function that quantifies the wave-like behavior, and $P$ is the function that quantifies the particle-like behavior. Quantifiers $W$ and $P$ can be flexibly chosen in different regions throughout the whole experimental setup. For example, for the MZI depicted in Fig. 2, let us consider that it is possible to adjust the transmission ($T_1$) and reflection ($R_1$) coefficients of the first biased beam splitter (BBS$_1$), where $T_1^2 + R_1^2 = 1$. Consider, for instance, that BBS$_2$ is a balanced beam splitter (BS), i.e., $T_2 = R_2 = 1/\sqrt{2}$. One way of quantifying $W$, introduced in this context by Greenberger and Yasin [46], is through interferometric visibility (IV) $\mathcal{V}$, defined by the probabilities of detection $p_i$ in a certain detector $D_i$ ($i = 0, 1$) outside the MZI (after BS$_2$) as

$$\mathcal{V}_i = \frac{\max(p_i) - \min(p_i)}{\max(p_i) + \min(p_i)}. \tag{18}$$

Strictly speaking, it is possible to derive two expressions for $\mathcal{V}$, one for each of the detectors. The maximum and minimum probabilities arise for a different phase variation $\phi_E$. Thus, the definition of IV as a wave-like behavior quantifier lies in the phase sensitivity of probabilities outside the MZI, characterizing $\mathcal{V}$ as a wave-like behavior quantifier by retro-inference. Predictability, which quantifies particle-like behavior, is frequently expressed as

$$\mathcal{P} = |T_1^2 - R_1^2|. \tag{19}$$

This quantifier can be understood as an *a priori* estimate, i.e., there is more information about the photon's path when $T_1$ or $R_1$ is closer to 1, and less information when $(T, R)$ have similar values, independently of the counts obtained after the experiment is completed. Thus, the CR is presented as $\mathcal{V}_i^2 + \mathcal{P}^2 = 1$, where $W = \mathcal{V}_i^2$, $P = \mathcal{P}^2$, and $\beta = 1$. It is worth noting that the construction of this kind of CR in the quantitative version of BCP is based on *ad hoc* methods, i.e., the functions $W$ and $P$ can be constructed in many ways and depend on the configuration of the experimental apparatus.

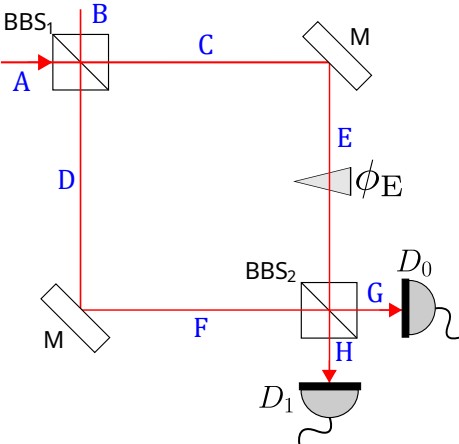

Figure 2: Mach-Zehnder interferometer composed of two BBSs. By utilizing a BBS$_j$ (with $j = 0, 1$), the amplitudes of transmission ($T_j$) and reflection ($R_j$) — which specify the probability amplitudes of the quantons in each MZI arm — can be controlled. Mirrors (M) convert the horizontal (vertical) spatial mode into the vertical (horizontal) mode while introducing a global phase factor of $e^{i\pi/2}$; the phase shifter applies a phase factor of $e^{i\phi_E}$ to the vertical spatial mode, and $D_0$ and $D_1$ represent the detectors.

The quantum version of BCP proposed in Ref. [32] and the traditional way to examine BCP differ in many ways. The former has a background supported by QM and gives us a formal definition for quantifying the QCR. A violation in this type of CR will reveal a fundamental issue with QM. In contrast, the latter still has a lack of conclusive formal definition, and there is great flexibility in the construction of complementarity relations, generally relying on the whole experimental apparatus along the lines of Bohr's view. For the particular case of MZI discussed above with BBS$_2$ = BS$_2$, the QCP and the BCP agree in the complementarity behavior of the quanton, but disagree in many situations as delineated in Ref. [32].

In order to develop the *ad hoc* question and introduce Afshar's experiment, another case is discussed to highlight the lack of a clear formal definition of BCP as discussed in Ref. [47]. In Fig. 2, when the BBS$_2$ represents a beam splitter (BS$_2$) by setting $T_2 = R_2 = 1/\sqrt{2}$, the visibilities $\mathcal{V}_0$ and $\mathcal{V}_1$ are the same and the BCP remains valid and agrees with QCP. In Ref. [47] the authors examined a more general configuration of the MZI where it is possible to vary the coefficients $T_2$ and $R_2$ of the second BBS, and a lack of a formal definition for BCP becomes clearly evident. This choice is completely admissible because there is no restriction for this in the literature. This setup shows that $\mathcal{V}_0$ and $\mathcal{V}_1$ are no longer equal, leading to situations where the quantitative BCP can be violated, highlighting its shortcomings and, in addition, showing that IV is not in general a good quantifier of $W$.

Afshar's experiment [29], depicted in Fig. 3, is a version of the double-slit experiment [48] that claims a violation of the BCP. Afshar used the *ad hoc* approach to the functions $W$ and $P$ combined with the lack of a clear definition of the BCP to construct a new CR in an attempt to understand the duality behavior in the double-slit experiment. The modified double-slit experiment is furnished with a wired grid depicted as black dots positioned in the $s_2$ plane. According to Afshar, the wire grid quantifies the maximum wave-like behavior through a non-destructive measurement, since the wires are positioned in regions of maximum destructive interference. Between the planes $s_2$ and $s_3$ there is a convex lens $L$ that redirects the beams to the detectors $D_A$ and $D_B$ positioned in the $s_3$ plane. In an alternative experimental configuration, closing the slit $A$ ($B$) results in only detector $D_B$ ($D_A$) recording clicks. Afshar *et al.* carried out the experiment while maintaining the wire grid in place and verified that the correlation between the path and the detector is maintained, with only a very small number of photons triggering the opposite detector; Afshar concluded that this mechanism of relating path and detector holds even when both slits are open. For more discussions about this path retro-inference approach, see Ref. [49]. When neither slit is blocked, wave-like characteristics are observable at $s_2$ (where $W$ reaches its peak) by the wire grid, and because only one detector will click for each photon, particle-like characteristics at $s_3$ denote the highest $P$. This scenario, according to Afshar, would represent a violation of the BCP, as there is a maximum $W$ and a maximum $P$ within the same experimental setup. It should be emphasized that there is extensive literature discussing Afshar's experiment [32, 49–58].

In the next section, we will analyze Afshar's experiment through the modified version of Unruh's experiment constructed through MZI.

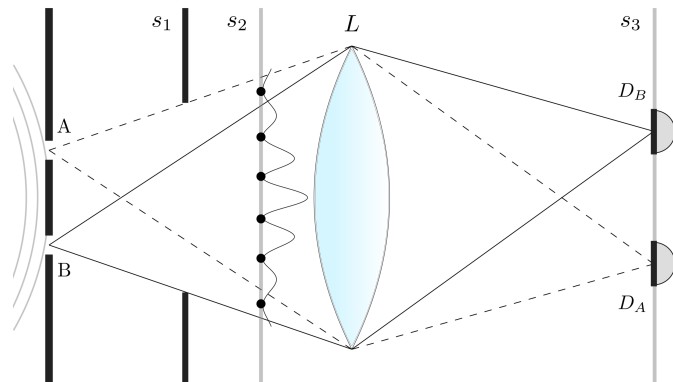

Figure 3: By modifying Young experiment and the methods of quantifying the wave- and particle-like behaviors of the quantons, Afshar claimed a violation of BCP. After measuring the positions of dark fringes (maximum destructive interference) at the screen $s_2$, Afshar replaced this screen by very thin wire grid at the positions previously determined. According to Afshar, this procedure is proposed for quantifying the wave-like behavior in a non-destructive way by verifying that this process does not significantly affect the photon counts at $s_3$. The area $s_3$ encompasses two positions, specifically detector $D_A$ and detector $D_B$, located beyond lens $L$, which direct the photons to these regions. In an auxiliary experiment, Afshar closes slit A (B) and verifies that only the corresponding detector $D_B$ ($D_A$) clicks; he concluded that, with both slits open, the slit-detector relation is still valid. With this in mind, since a maximum wave-like behavior is captured by the wire grid and a maximum particle-like behavior is obtained by the slit-detector relation, the author concludes that the BCP is violated.

### C. Modified Unruh's experiment

Unruh's experiment [30] was originally proposed as a two-level system analogy to Afshar's experiment to examine whether or not BCP is violated. More recently, Ref. [32] examined this experiment to demonstrate the applicability of the QCP and its effectiveness in addressing duality behavior through QCRs.

Pessoa Júnior's modified version of Unruh's experiment [31], depicted in Fig. 4, introduces two additional phase shifters: $\phi_{\rm E}$ in MZI$_1$ and $\phi_{\rm H}$ in MZI$_2$. His motivation for incorporating these phase shifts is to better align Unruh's interferometric experiment with the time-step analysis of Afshar's experiment. The original Unruh's setup, depicted in Fig. 4, corresponds to the case where $\phi_{\rm E} = \phi_{\rm H} = 0$.

As discussed in Sec. II B, Afshar employed an *ad hoc* approach to define wave and particle properties in a double-slit experiment, claiming that his findings constituted a violation of the quantitative version of BCP. In the modified Unruh setup, MZI$_2$ serves as an analog to the wire grid region in Afshar's experiment, where mode H represents the minima of the interference pattern, capturing wave behavior non-destructively. In the absence of the blocker $B_0$, the state after BS$_2$ exhibits complete destructive interference in mode H and fully constructive interference in mode G. Unruh noted that mode H corresponds to the dark fringes where the wires were placed in Afshar's experiment to probe wave-like behavior. Importantly, the presence or absence of blocker $B_1$ in mode H does not alter this wave-like behavior when both slits are opened. When $B_1$ is absent, selectively inserting $B_0$ into mode C or D allows one to associate a detection event at $D_1$ with a quanton traveling via the upper path (modes C and E) and a detection at $D_0$ with the lower path (modes D and F). Following Afshar's interpretation, a click at either detector in the absence of $B_0$ is taken as an indication of the quanton's path, suggesting maximal corpuscular behavior.

A key objection to this reasoning arises when $B_1$ is placed in mode H after use $B_0$ to block one of the paths. In this case, the previous path-detector association (established by blocking C or D with $B_0$) is no longer valid, as the interference structure is altered. Afshar *et al.* conducted their experiment with the wire grid in place and observed that the opposing detector registered approximately 0.5% of photons, preserving the path-detector relationship. This stands in stark contrast to Unruh's version, where the presence of $B_1$ completely disrupts the path-detector correlation.

Let us begin the formal analysis considering the phases $\phi_{\rm E}$ and $\phi_{\rm H}$ and subsequently taking into account the observations of Pessoa Júnior. We consider the initial state $|\psi_0\rangle = |1_{\rm A}, 0_{\rm B}\rangle = |10\rangle_{\rm AB}$. After BS$_1$, the state evolves to $|\psi_1\rangle = \frac{1}{\sqrt{2}}\big(|10\rangle_{\rm CD} + i|01\rangle_{\rm CD}\big)$. The mirrors (M) change the mode from horizontal (vertical) to vertical (horizontal) and add a phase of $e^{i\pi/2} = i$. The phase shifter adds a phase of $e^{i\phi_{\rm E}}$ when mode E is populated. The state after these two optical elements and BS$_2$ produces $|\psi_2\rangle = -\left(\frac{e^{i\phi_{\rm E}}+1}{2}\right)|10\rangle_{\rm GH} + i\left(\frac{e^{i\phi_{\rm E}}-1}{2}\right)|01\rangle_{\rm GH}$. Finally, after phase $e^{i\phi_{\rm H}}$, mirrors and BS$_3$, the state evolves to

$$|\psi_3\rangle = -\frac{1}{2\sqrt{2}}\left(e^{i\phi_{\rm E}}e^{i\phi_{\rm H}} - e^{i\phi_{\rm H}} - e^{i\phi_{\rm E}} - 1\right)|10\rangle_{\rm KL} - \frac{i}{2\sqrt{2}}\left(e^{i\phi_{\rm E}}e^{i\phi_{\rm H}} + e^{i\phi_{\rm E}} - e^{i\phi_{\rm H}} + 1\right)|01\rangle_{\rm KL}. \tag{20}$$

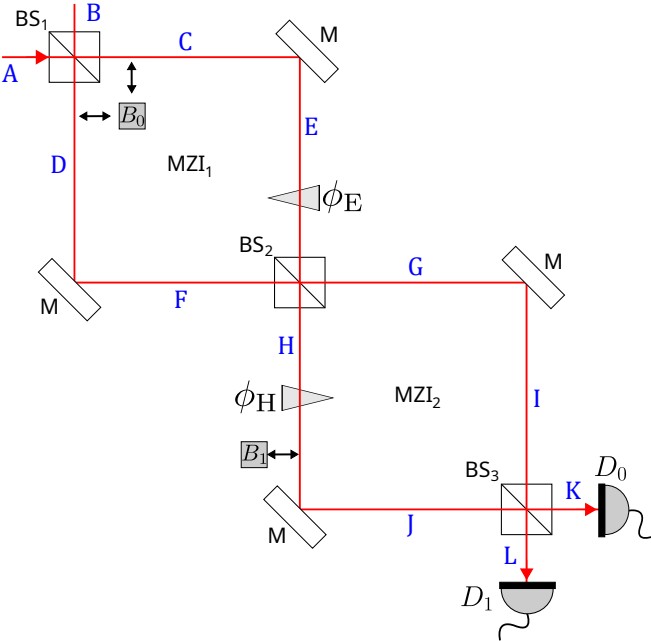

Figure 4: Modified Unruh's experiment proposed by Pessoa Júnior in Ref. [31]. The setup is composed of an extra MZI, namely MZI$_2$, inserted subsequently to the MZI$_1$ output modes. The main difference in this proposed version is choosing appropriate values for the phase shifts $\phi_\mathrm{E}$ and $\phi_\mathrm{H}$ to align Unruh's experiment with the original aim of Afshar's experiment. The boxes $B_0$ and $B_1$ represent the blockers that can be inserted into the quanton paths.

Pessoa Júnior explored the introduction of two phase shifts, $\phi_\mathrm{E}$ and $\phi_\mathrm{H}$, selecting a specific case to demonstrate how it aligns more closely with Afshar's experiment. Since the Greenberger-Yasin formulation of the BCP does not impose explicit restrictions on the use of IV, he follows the usual literature and assumes that Eq. (18) remains valid for MZI$_1$ even after BS$_3$.

According to Pessoa Júnior, in Afshar's experiment one can examine different regions to interpret the quanton's behavior at each time step. Immediately after BS$_1$ (in MZI$_1$), the behavior is corpuscular. This notion, borrowed from the literature—particularly from experiments such as Wheeler's delayed-choice (WDCE) [59]—suggests that a click in a detector following a photon's passage through the upper slit (upper arm of the MZI) or lower slit (lower arm of the MZI) implies that the photon originated from that specific path. Inside MZI$_2$, the quanton's behavior is wave-like, as indicated by IV in Eq. (18). This aligns with the role of the wire grid region in Afshar's experiment, where the wave nature of light is inferred. The phase settings that capture this sequence of behaviors in Pessoa Júnior's modified version of Unruh's experiment are $\phi_\mathrm{E} = \pi/2$ and $\phi_\mathrm{H} = 0$.

By keeping $\phi_\mathrm{H} = 0$ and varying $\phi_\mathrm{E}$, no wave-like behavior emerges through IV, and the probabilities in Eq. (20) remain unchanged for different values of $\phi_\mathrm{E}$, characterizing an absence of wave-like behavior in MZI$_1$. Conversely, fixing $\phi_\mathrm{E} = \pi/2$ while varying $\phi_\mathrm{H}$ maximizes the wave-like behavior in MZI$_2$, as inferred from the IV. In this case, for $\phi_\mathrm{E} = \pi/2$, the probabilities in Eq. (20) become sensitive to variations in $\phi_\mathrm{H}$, capturing the wave phenomenon in MZI$_2$. After BS$_3$, the behavior becomes again corpuscular, following the same reasoning as after BS$_1$. Pessoa Júnior argued that these phase choices effectively represent each step of Afshar's setup. However, he also noted unresolved issues concerning the inclusion of the wire grid in the interferometric version. This motivated his proposal for the experiment explored in Sec. II D.

The standard MZI is well known for its instability in optical device applications. As a result, the Sagnac interferometer [60, 61] is often preferred in experimental settings due to its greater stability. However, to maintain the original structure of the experiments and ensure consistency with the original analysis of the CR, the simulation was performed on quantum computers using the formalism outlined in Sec. II A. While the modified Unruh's experiment and the alternative version of the Pessoa Júnior experiment discussed in the next Section do not require second quantization if we do not consider the blockers, the inclusion of blockers introduces a less straightforward situation. To overcome these difficulties and align the experiment to something that is more in line with what would occur in a bench experiment, we employ the formalism used in Refs. [9–11]. Thus, we effectively captured the scenario of "absorption" in the blocker; when absorption is absent, the experiment proceeds unchanged, allowing both scenarios to be represented within a single quantum circuit. Therefore, a key advantage of this formalism in the present context is its ability to facilitate a clearer analysis of the role of blockers through quantum computer simulations.

*1. Digital quantum simulation of the modified Unruh's experiment*

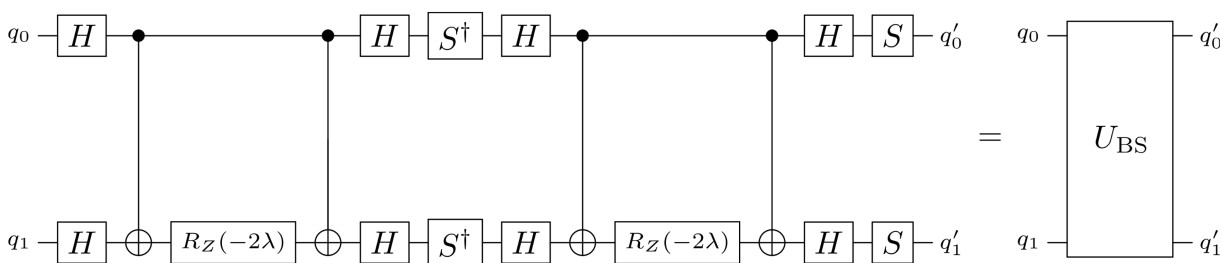

Figure 5: Quantum circuit for a BS, for which it is possible to control the transmission, $T$, and the reflection, $R$, coefficients with $\lambda = \arctan\left(\frac{R}{T}\right)/2$. Each qubit represents one of the input modes. Following the execution of the circuit, the qubits that represent the input modes $q_0, q_1$ are converted into the output mode $q_0', q_1'$. For example, in Fig. 1 the input modes A ($q_0$) and B ($q_1$) are converted into the output modes C ($q_0'$) and D ($q_1'$). The entire quantum circuit is represented by the box $U_{\mathrm{BS}}$.

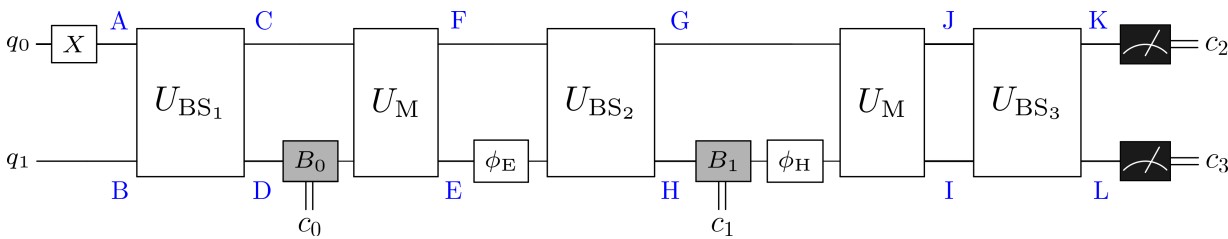

Figure 6: Quantum circuit for the modified Unruh's experiment. The state of qubits $q_0$ and $q_1$ correspond to the states of horizontal (A, C, F, G, J, K) and vertical (B, D, E, H, I, L) modes, respectively. The initial state prior to $U_{\mathrm{BS}_1}$ is $|\psi_0\rangle = |1\rangle_{\mathrm{A}} |0\rangle_{\mathrm{B}}$, with the X gate required to initiate mode A as occupied. The double lines denoted by $c_0$ and $c_1$ represent the classical bits storing the results registered by the blockers $B_0$ and $B_1$, respectively, and $c_2$ and $c_3$ by the detectors $D_0$ and $D_1$, respectively.

To simulate the modified Unruh's experiment, depicted in Figure 4, we follow the procedure outline in Section II A. In the context of these experiments, the maximum occupation number of each mode is $n_j = 1 \; \forall j$, and the total occupation number $N$ is also equal to 1. To obtain the action of $U_{\mathrm{BS}}$ in states with this restriction, we first evolve the operators in the Heisenberg picture:

$$|1_{\mathrm{A}}, 0_{\mathrm{B}}\rangle = a^{\dagger} |0_{\mathrm{A}}, 0_{\mathrm{B}}\rangle = \left(Tc^{\dagger} + iRd^{\dagger}\right)|0_{\mathrm{C}}, 0_{\mathrm{D}}\rangle = T |1_{\mathrm{C}}, 0_{\mathrm{D}}\rangle + iR |0_{\mathrm{C}}, 1_{\mathrm{D}}\rangle, \tag{21}$$

$$|0_{\mathrm{A}}, 1_{\mathrm{B}}\rangle = b^{\dagger} |0_{\mathrm{A}}, 0_{\mathrm{B}}\rangle = \left(iRc^{\dagger} + Td^{\dagger}\right)|0_{\mathrm{C}}, 0_{\mathrm{D}}\rangle = iR |1_{\mathrm{C}}, 0_{\mathrm{D}}\rangle + T |0_{\mathrm{C}}, 1_{\mathrm{D}}\rangle. \tag{22}$$

The action of a balanced BS ($T = R = 1/\sqrt{2}$) on these states is then given (in the Schrödinger picture) by

$$|1_{\mathrm{A}}, 0_{\mathrm{B}}\rangle \xrightarrow{U_{\mathrm{BS}}} \frac{1}{\sqrt{2}}\left(|1_{\mathrm{C}}, 0_{\mathrm{D}}\rangle + i |0_{\mathrm{C}}, 1_{\mathrm{D}}\rangle\right), \tag{23}$$

$$|0_{\mathrm{A}}, 1_{\mathrm{B}}\rangle \xrightarrow{U_{\mathrm{BS}}} \frac{1}{\sqrt{2}}\left(i |1_{\mathrm{C}}, 0_{\mathrm{D}}\rangle + |0_{\mathrm{C}}, 1_{\mathrm{D}}\rangle\right), \tag{24}$$

which are well-known superposition states for a single photon passing through a BS. The next step is mapping these states to qubit states. Since the focus is on representing occupation numbers 0 and 1, and $U_{\mathrm{BS}}$ acts on a total of two modes, the map can be written as

$$|n_{\mathrm{A}}, \; n_{\mathrm{B}}\rangle_F \; \mapsto \; |n_{\mathrm{A}}\rangle \otimes |n_{\mathrm{B}}\rangle, \tag{25}$$

with $n_{\mathrm{A}}, n_{\mathrm{B}} \in \{0, 1\}$ such that $n_{\mathrm{A}} + n_{\mathrm{B}} = 1$. If the system were composed of $N \geq 2$ particles, codification methods (e.g. decimal to Gray code, discussed in Sec. II A) would be necessary to represent the numbers $n_{\mathrm{A}}$ and $n_{\mathrm{B}}$; in our case, it is not necessary since 0 and 1 have the same representation in decimal, binary and Gray codes. Consequently,

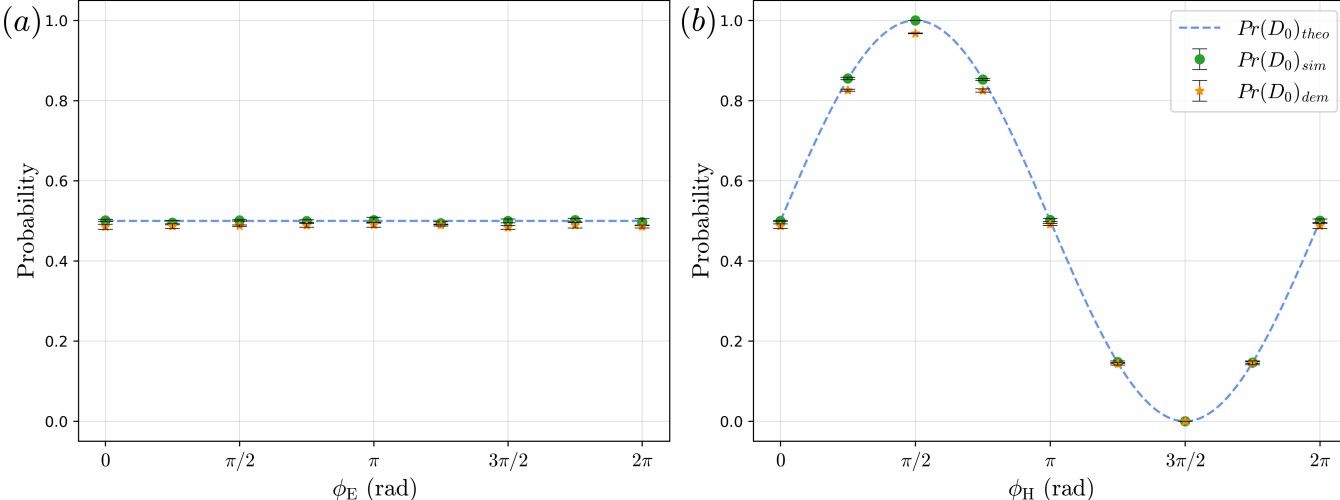

Figure 7: The graphs above illustrate the changes in probability at detector $D_0$ as a function of one of the phase shifts, while the other is fixed. (a) Probabilities for a variation of $\phi_{\mathrm{E}}$, with $\phi_{\mathrm{H}} = 0$ fixed. The interferometric visibility is theoretically zero, because the maximum and minimum probabilities are equal. This indicates a the absence of wave-like behavior at $\mathrm{MZI}_1$, as the phase variation does not produce changes in probability. (b) Probabilities for a variation of $\phi_{\mathrm{H}}$, with $\phi_{\mathrm{E}} = \pi/2$ fixed. This is interpreted as a wave-like behavior at $\mathrm{MZI}_2$, as the phase variation now produces changes in probability, with $\max Pr(D_0) = 1$ and $\min Pr(D_0) = 0$. The error bars correspond to the standard deviation for five repetitions of the classical simulations (circle markers) and quantum demonstrations (star markers) using IBMQ's Heron r2 processor ibm_kingston quantum chip (its calibration parameters are given in the supplemental material [63])

.

the map between the bosonic operators and Pauli operators is given by

$$a^\dagger |0_{\mathrm{A}}, 0_{\mathrm{B}}\rangle = |1_{\mathrm{A}}, 0_{\mathrm{B}}\rangle \mapsto (\mathcal{S}_1 \otimes \mathbb{I}) |0_{\mathrm{A}}\rangle |0_{\mathrm{B}}\rangle = |1_{\mathrm{A}}\rangle |0_{\mathrm{B}}\rangle, \tag{26}$$

$$a |1_{\mathrm{A}}, 0_{\mathrm{B}}\rangle = |0_{\mathrm{A}}, 0_{\mathrm{B}}\rangle \mapsto (\mathcal{S}_0 \otimes \mathbb{I}) |1_{\mathrm{A}}\rangle |0_{\mathrm{B}}\rangle = |0_{\mathrm{A}}\rangle |0_{\mathrm{B}}\rangle, \tag{27}$$

$$b^\dagger |0_{\mathrm{A}}, 0_{\mathrm{B}}\rangle = |0_{\mathrm{A}}, 1_{\mathrm{B}}\rangle \mapsto (\mathbb{I} \otimes \mathcal{S}_1) |0_{\mathrm{A}}\rangle |0_{\mathrm{B}}\rangle = |0_{\mathrm{A}}\rangle |1_{\mathrm{A}}\rangle, \tag{28}$$

$$b |0_{\mathrm{A}}, 1_{\mathrm{B}}\rangle = |0_{\mathrm{A}}, 0_{\mathrm{B}}\rangle \mapsto (\mathbb{I} \otimes \mathcal{S}_0) |0_{\mathrm{A}}\rangle |1_{\mathrm{B}}\rangle = |0_{\mathrm{A}}\rangle |0_{\mathrm{B}}\rangle. \tag{29}$$

The map between the two-site bosonic interaction operators is thus $b^\dagger a + b a^\dagger \mapsto \frac{1}{2}(X \otimes X + Y \otimes Y)$ and the unitary for the BS, presented in Eq. (1), is given by $U_{\mathrm{BS}} = e^{i\lambda(X \otimes X + Y \otimes Y)}$, where $\lambda = \theta/2$. Given that $[X \otimes X, Y \otimes Y] = \mathbb{0}$, it follows that

$$U_{\mathrm{BS}} = e^{i\lambda(X \otimes X)} e^{i\lambda(Y \otimes Y)}. \tag{30}$$

Figure 5 illustrates the quantum circuit for simulating a BS for any $\theta$. In Appendix B, a complete derivation of the unitary form for a BS is provided, along with instructions on how to build the corresponding quantum circuit.

To construct the quantum circuit for the modified Unruh's experiment, we begin by setting the input modes to $q_0$ (A) and $q_1$ (B) and the output modes to $q'_0$ (C) and $q'_1$ (D). As all qubits are initialized in the state $|0\rangle$ by default, to prepare the initial state $|\psi_0\rangle = |1_{\mathrm{A}}, 0_{\mathrm{B}}\rangle \mapsto |10\rangle_{\mathrm{AB}}$, one can apply the quantum gate $X = |0\rangle\langle 1| + |1\rangle\langle 0|$ to qubit A to change it into an occupied mode. To transform the quantum circuit in Fig. 5 into a quantum circuit for a balanced BS it is necessary to set $\lambda = \pi/8$. Mirrors are also a particular case of the circuit in Fig. 5 for $\lambda = \pi/4$. The phase shifter is represented by the unitary $P(\phi_{\mathrm{mode}}) = |0\rangle\langle 0| + e^{i\phi_{\mathrm{mode}}} |1\rangle\langle 1|$, i.e., the term $e^{i\phi_{\mathrm{mode}}}$ is only applied if the mode is occupied. The parallel double lines represent the storage of the classical bits registered by the blockers and the detectors. Figure 6 shows the complete quantum circuit for the modified Unruh's experiment.

In optical experiments, blockers absorb the photons and end the experiment, but in quantum computers there are no blockers to implement intrinsically. The solution is not straightforward in the usual quantum simulation of two-level systems, but it is very simple in the second quantization approach and consists of a measuring device followed by a resetting device, which enables the blocking simulation and the measurements after $\mathrm{BS}_3$ in a single setup. The resetting device converts any state to $|0\rangle$. Thus, a measurement device is inserted to get the result for that mode, and the resetting device turns the mode to the non-occupied state. If the mode is in the state $|0\rangle$, this means that

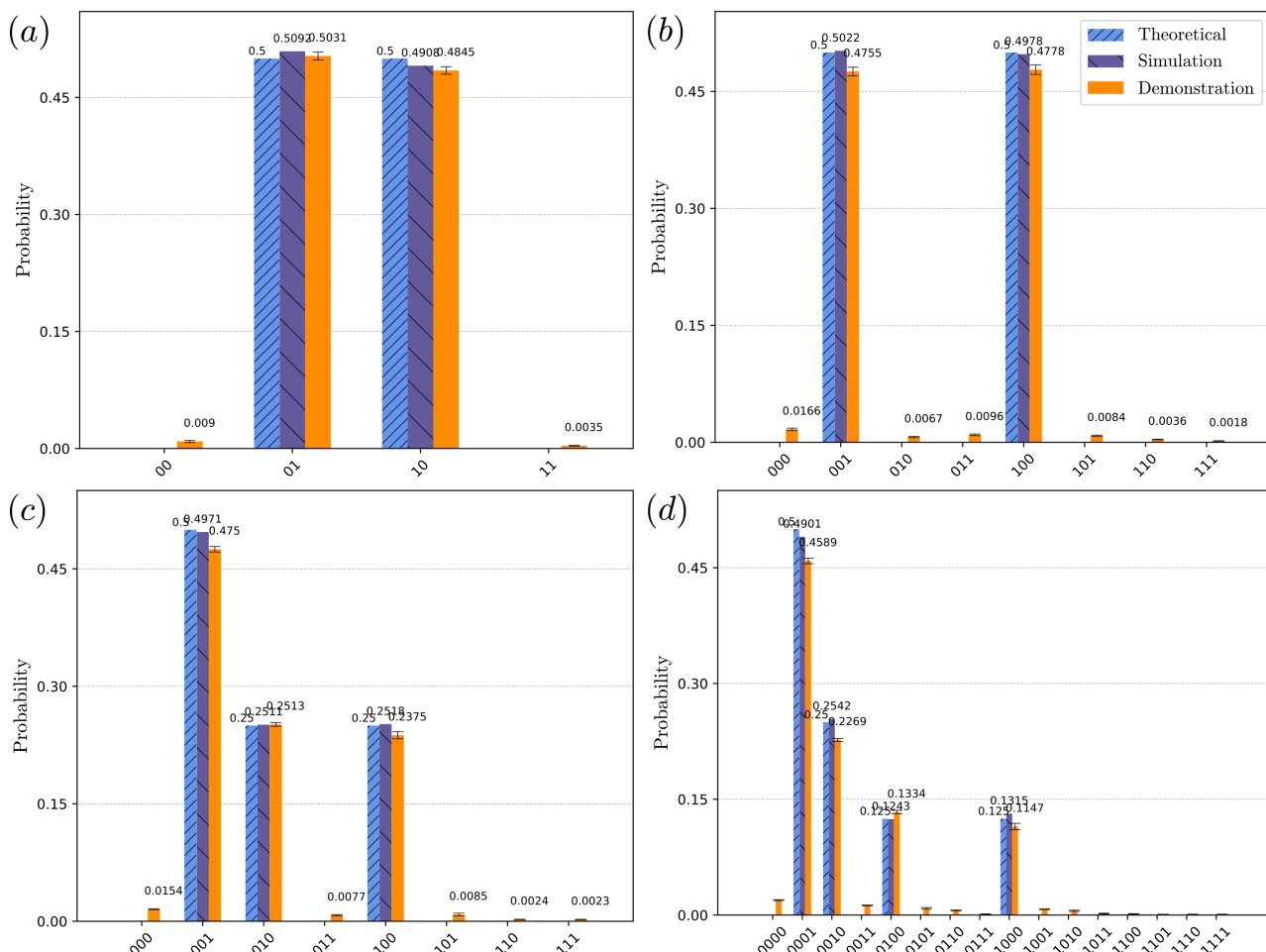

Figure 8: Histograms depicting the theoretical (bars with very close diagonal hatching), simulation (bars with widely spaced diagonal hatching), and demonstration (solid bars) results for four different configurations of the modified Unruh's experiment circuit: (a) no blocker inserted and classical bits composed by $c_3 c_2$. The photons are detected either in $D_0$ (result 01) or $D_1$ (result 10). (b) Blocker inserted in mode D and classical bits composed by $c_3 c_2 c_0$. About 50% of the photons hit the blocker $B_0$ (result 001) and the other 50% pass through the upper arm (mode C) and are detected in $D_1$ (result 100). (c) Inserting only the blocker $B_1$ in mode H and classical bits composed by $c_3 c_2 c_1$; this mode is no longer in the vacuum state as in the original Unruh's experiment. About 50% of the photons hit the blocker $B_1$ (result 001) and 25% are detected in $D_0$ (result 010) and $D_1$ (result 100). (d) Both blockers inserted (modes D and H) and classical bits composed by $c_3 c_2 c_1 c_0$. About half the photons hit the first blocker (result 0001), and about half of those which do not (25%) end up hitting the second blocker (result 0010). The photons that are not absorbed by either blockers are detected in both $D_0$ (result 0100) and $D_1$ (result 1000), 12.5% each. The error bars correspond to the standard deviation for five repetitions of the quantum demonstrations, which were obtained with IBMQ's Heron r2 processor ibm_kingston quantum chip. The complete results (including those omitted to improve visual clarity) and the chip's calibration parameters can be found in the supplemental material [63].

there is no photon to block, the experiment continues and the reset device has no practical effect. In contrast, if the mode is in the state $|1\rangle$, the state should be set to $|0\rangle$, indicating that the photon was absorbed by the blocker. The experiment continues with a measurement after $BS_3$, analogous to the case in which the photon is not blocked.

Figure 7 presents the graphs of probabilities for the phases $\phi_E = \pi/2$ and $\phi_H = 0$ considered by Pessoa Júnior. In Fig. 7 (a), $\phi_H = 0$ is fixed while $\phi_E \in [0, 2\pi]$. Theoretically, the visibility of Eq. (18) in $MZI_1$ is zero, characterizing the absence of wave-like behavior. In Fig. 7 (b), the phase $\phi_E = \pi/2$ is fixed while $\phi_H \in [0, 2\pi]$. Theoretically, the visibility of Eq. (18) in $MZI_2$ equals one, characterizing wave-like behavior. In both graphs, the dotted blue lines represent the theoretical probabilities $(Pr(D_0)_{theo})$, the circular green markers represent the classical simulation results $(Pr(D_0)_{sim})$, and the orange star markers represent the demonstrative results $(Pr(D_0)_{dem})$ obtained using IBMQ's Heron r2 processor ibm_kingston quantum chip (its calibration parameters are given in the supplemental material [63]). Despite the fact that the effects of noise and hardware constraints on quantum devices are well acknowledged and impact the final results, the demonstrative results are in reasonable agreement with theoretical

predictions and classical simulations.

Before the analysis of the results in the histograms in Fig. 8, consider the following description: $i$) 8192 measurements were carried out for each demonstrative result, each producing a sequence of classical bits. $ii$) Fig. 6 indicates which measurement result is associated with each $c_j$, for $j = 0, 1, 2, 3$: $c_0$ is associated with the inclusions of blocker $B_0$, $c_1$ with blocker $B_1$, $c_2$ with detector $D_0$ and $c_3$ with detector $D_1$. $iii$) The binary numbers shown in the histograms are derived from measurements corresponding to a descending index of $c_j$. $iv$) Each measurement procedure yields as output a bit string, where 0's represent non-occupied modes and 1's represent occupied modes. Ideally, in this case, only one position in the bit string has a non-zero bit.

The histogram in Fig. 8 ($a$) shows the case without blockers, where the sequence 01 (10) $= c_3 c_2$ means that about 50% of the photons are detected in $D_0$ ($D_1$). In Fig. 8 ($b$) the blocker $B_0$ is inserted in qubit $q_1$ (mode D); about 50% of the photons hit this blocker, producing the binary sequence 001 $= c_3 c_2 c_0$. The sequence 100 $= c_3 c_2 c_0$ means that about 50% of the detections occur in the detector $D_1$. According to Afshar, a detection in $D_1$ can be correlated to the upper path (mode C). Although the simulation with the blocker $B_0$ in mode C is not presented, the sequence 010 would correspond to a detection in $D_0$, which correlates to the lower path (mode D). Fig. 8 ($c$) shows the case where both paths are free and blocker $B_1$ is inserted in mode H. The binary number 001 $= c_3 c_2 c_1$ means that about 50% of photons hit this blocker; the binary number 010 $= c_3 c_2 c_1$ (100 $= c_3 c_2 c_1$) means that the photons that passed through were detected in $D_0$ ($D_1$). This case differs from the original approach by Unruh, in the sense that mode H is not exclusively in the vacuum state anymore. Fig. 8 ($d$) shows the case with both blockers in place. About 50% of the photons hit blocker $B_0$ (0001 $= c_3 c_2 c_1 c_0$), and about 25% hit the blocker $B_1$ (0010 $= c_3 c_2 c_1 c_0$). The results 0100 $= c_3 c_2 c_1 c_0$ and 1000 $= c_3 c_2 c_1 c_0$ represent detections in $D_0$ and $D_1$, respectively. According to Pessoa Júnior, when modes C and D are free, the inclusion of $B_1$ is no longer possible since after the inclusion of $B_0$ the correlation between one path and one detector is no longer possible. In the following subsection, the experimental arrangement proposed by Pessoa Júnior is examined to demonstrate that the path-detector correlation still holds when $B_1$ is placed, similar to the presence of the wire grid in Afshar's experiment.

## D.   Pessoa Júnior's experiment - an enhancement of the modified Unruh's experiment

As shown in Sec. II C, Pessoa Júnior [31] explores a modified version of Unruh's experiment to better capture the behavior of the quanton in each step of Afshar's experiment by adjusting phase shifts in $\text{MZI}_1$ and $\text{MZI}_2$. However, when modes C and D are free, mode H, which was in the vacuum state in the version presented by Unruh, is no longer so. Moreover, the loss of photons by the insertion of the wire grid while each of the slits is blocked individually is not completely integrated into the modified Unruh's experiment. In view of this, Pessoa Júnior proposed another experimental apparatus, depicted in Fig. 9. It consists of replacing the mirrors in the $\text{MZI}_1$ of Fig. 4 for BBS's (upper mirror for $\text{BBS}_4$ and lower mirror for $\text{BBS}_5$).

The discussion presented by Pessoa Júnior is deviated from at this point, and the experimental arrangement is adjusted to replicate certain aspects of Afshar's experiment. The same sequence presented in Fig. 8 is followed, and $\phi_H = \pi/2$ is included. The transmissibility of the BBS's were set to $T_4^2 = T_5^2 =: T^2 = 0.96$ to simulate the loss of photons when the slits are open and the wire grid is inserted. The higher transmissibility can be understood as the region between the wire grids that allows the free flow of photons without blocking. The value of $T$ was chosen in order to obtain the 2% probability of quantons hitting the blocker $B_1$ which, in analogy to Afshar's experiment, is close to the amount of photons lost in that situation. With one arm blocked by $B_0$ and with $B_1$ in place, the probability of detection at $D_1$ is around 0.5%, which simulates photons that arrive at the opposing detector due to the inclusion of the wire grid.

The fact that Pessoa Júnior's setup includes two additional beam splitters in the first MZI results in the necessity of including two additional input modes (E and F) into the description of this system. Modes G and M (J and P), as well as the mode Q (R), lead to detector $D_0$ ($D_1$). Unlike in the modified Unruh's experiment, detector $D_0$ ($D_1$) is related to path C (D) and the inclusion of $\phi_N = \pi$ in mode N is made to redirect the photons of modes C and D to the corresponding detector. In the formal analysis, as before, the initial state is given by $|\psi_0\rangle = |1_A 0_B 0_E 0_F\rangle \equiv |1000\rangle_{ABEF}$ in which mode A is occupied by one photon. After $\text{BS}_3$, the state is given by $|\psi\rangle_{MQRP} = \frac{i}{\sqrt{2}} (|D_0\rangle + |D_1\rangle)$ where $|D_0\rangle = T |1000\rangle_{MQRP} - R |0100\rangle_{MQRP}$ and $|D_1\rangle = iT |0001\rangle_{MQRP} + R |0010\rangle_{MQRP}$. The complete state evolution can be found in Appendix C. In the absence of a blocker, the probability that a photon reaches detector $D_0$ or $D_1$ is 1/2.

Under the condition that blockers are present, the interaction-free measurement procedure introduced by Elitzur and Vaidman in Ref. [62] is employed in the description of the quantum states. The procedure for constructing states follows the same structure as described without blockers in Appendix C. The key is that the quantum state, which does not interact with the blocker, continues its unitary evolution as if it were a newly prepared state in the respective mode. With $B_0$ placed in mode D (the same approach can be used for $B_0$ in mode C), the state is given

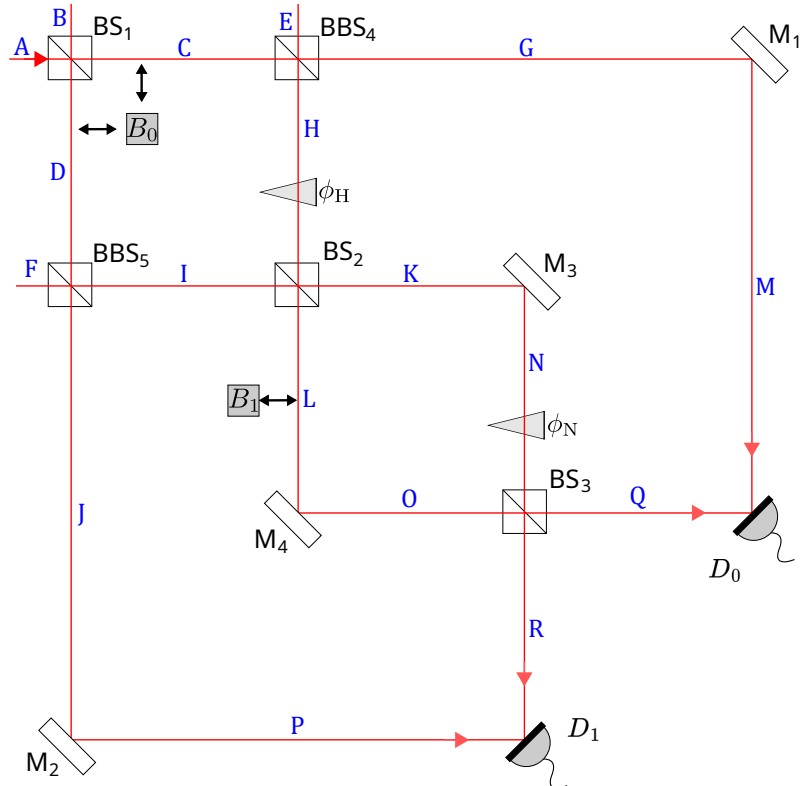

Figure 9: Pessoa Júnior's experiment is designed through examining the extra phase shifters incorporated into Unruh's model. In $\text{MZI}_1$ shown in Fig. 4, the mirrors were substituted by BBSs, namely $\text{BBS}_4$ and $\text{BBS}_5$ in the upper and lower arms of the interferometer, respectively. As discussed previously, Unruh's configuration blocks all the photons that pass through $B_1$, destroying the correlation between the path in $\text{MZI}_1$ and the detectors. The BBS's introduce two new input modes, E and F, in the description of the quantum states. The transmissibility in $\text{BBS}_4$ and $\text{BBS}_5$ is adjusted to solve the problem introduced in Unruh's experiment after the inclusion of blocker $B_1$ and to align it with Afshar's experiment, where the relation between path and detector is made with the wire grid in place. The phase shift in mode H is set to $\phi_\text{H} = \pi/2$, as in the previous experimental setup; $\phi_\text{N} = \pi$ is used so that only one detector will trigger when blocking modes C or D.

by $|\psi\rangle_\text{DMQR} = \frac{i}{\sqrt{2}} (|B_0\rangle + |D_0\rangle)$, where $|B_0\rangle = |1000\rangle_\text{DMQR}$ represents the quantum state of the photon just before interacting with the blocker $B_0$ and $|D_0\rangle = T|0100\rangle_\text{DMQR} - R|0010\rangle_\text{DMQR}$ represents the quantum state of the photon just before reaching the detector $D_0$. The probability of the photon being blocked or being detected by $D_0$ is 1/2. The path of the relation C and the detector $D_0$ is maintained following Afshar.

With $B_1$ in place, the state is given by $|\psi\rangle_\text{LMQRP} = |B_1\rangle + |D_0\rangle + |D_1\rangle$ where $|B_1\rangle = \frac{iR}{2}(i-1)|10000\rangle_\text{LMQRP}$, $|D_0\rangle = \frac{iT}{\sqrt{2}}|01000\rangle_\text{LMQRP} - \frac{R(1-i)}{2\sqrt{2}}|00100\rangle_\text{LMQRP}$, and $|D_1\rangle = -\frac{T}{\sqrt{2}}|00001\rangle_\text{LMQRP} - \frac{R(1+i)}{2\sqrt{2}}|00010\rangle_\text{LMQRP}$. Afshar *et al.* observed a photon loss of around 2% when both slits were simultaneously open and the wire grid was positioned in place. So, by setting $T^2 = 0.96$, the probability that the quanton hits $B_1$ is around the same as the percentage of photon loss reported by those authors. The probability of quantons being detected in $D_0$ or $D_1$ is 49%.

Finally, with $B_0$ in mode D and $B_1$ in place, the state is given by $|\psi\rangle_\text{DLMQR} = |B_0\rangle + |B_1\rangle + |D_0\rangle + |D_1\rangle$ where $|B_0\rangle = \frac{i}{\sqrt{2}}|10000\rangle_\text{DLMQR}$, $|B_1\rangle = -\frac{R}{2}|01000\rangle_\text{DLMQR}$, $|D_0\rangle = \frac{i}{\sqrt{2}}T|00100\rangle_\text{DLMQR} - \frac{iR}{2\sqrt{2}}|00010\rangle_\text{DLMQR}$, and $|D_1\rangle = -\frac{R}{2\sqrt{2}}|00001\rangle_\text{DLMQR}$. The probability of the quanton being blocked by $B_0$ is 50%, and by $B_1$, 1%; the probability of the quanton being detected in $D_0$ (the corresponding detector of mode C) is 48.5%, and in the opposing detector $D_1$, 0.5%. The latter is proportionally greater than that reported in Afshar's experiment, but highlights that the inclusion of $B_1$ affects the path-detector relation. However, this experimental setup fails to simulate the greater loss of photon count caused by the presence of the wire grid, which is reported to be approximately 14% in the original experiment.

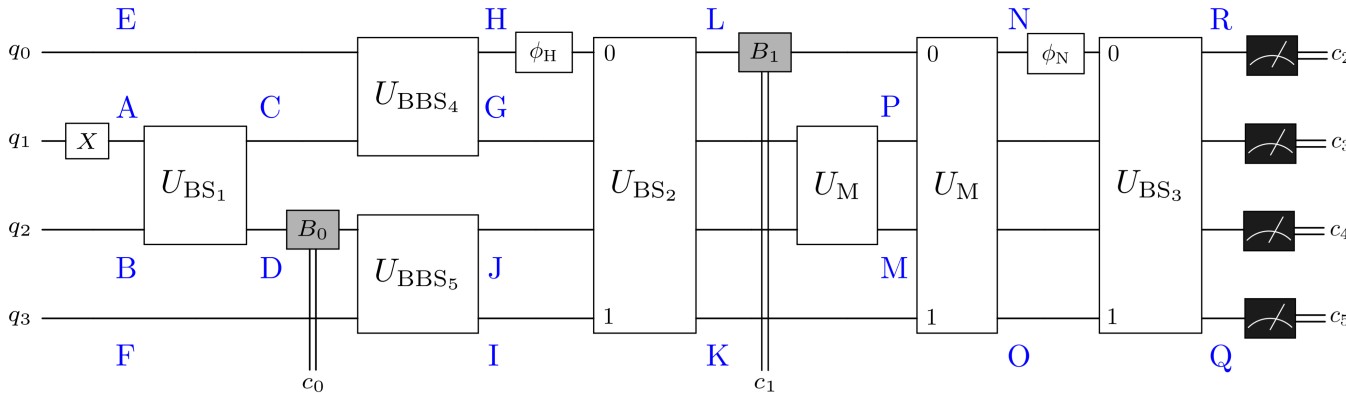

Figure 10: Quantum circuit for simulating the modified Pessoa Júnior's experiment. The state of qubits $q_1$ and $q_3$ ($q_0$ and $q_2$) correspond to the states of horizontal (vertical) modes, with the initial state before $U_{BS_1}$ being $|\psi_0\rangle = |0\rangle_E |1\rangle_A |0\rangle_B |0\rangle_F$. In comparison to the previous setup, this version has two additional input modes E and F which arise from the inclusion of $U_{BBS_4}$ and $U_{BS_5}$. The double lines denoted by $c_0$ and $c_1$ represent the classical bits storing the results registered by the blockers. The results of measuring qubits $q_0$ and $q_1$ ($q_2$ and $q_3$) are stored in $c_2$ and $c_3$ ($c_4$ and $c_5$), which correspond to the detections made by detector $D_1$ ($D_0$) as depicted in Fig. 9. The small numbers $(0, 1)$ adjacent to the multi-qubit gates indicate the qubits on which each gate acts; for example, in the operation $U_{BS_2}$ only the qubits in modes H and I are involved.

### 1. Digital simulation of Pessoa Júnior's experiment

The digital simulation of the experimental version proposed by Pessoa Júnior follows the same idea as that constructed in Sec. II C 1 and Eqs. (29)-(30) are the same. The major difference from the previous setup is the exchange of MZI$_1$ mirrors for BBS's. There are also two new input modes (E and F, both in the vacuum state) represented by the qubits $q_0$ and $q_3$, as depicted in Fig. 10. As before, the initial state is $|\psi_0\rangle = |1000\rangle_{ABEF}$, which is produced by applying the Pauli $X$ gate on qubit $q_1$. This state is first evolved by $U_{BS_1}$ and then by $U_{BSS_4}$ and $U_{BSS_5}$, which receive as input the modes C, E and D, F, respectively. The output modes H (after the phase shifter $\phi_H = \pi/2$) and I are then the input modes of $U_{BS_2}$. The unitary representing the innermost mirrors act on modes K and L leading to the input modes of $U_{BS_3}$ (N, after $\phi_N = \pi$, and O), which in turn has output modes Q and R. The unitary representing the outermost mirrors act on modes G and J, leading to the remaining final output modes, M and P. Finally, quantons from modes M and Q are detected in $D_0$, whereas those from P and R are detected in $D_1$. The measurement results are encoded in classical bits: $c_0$ is assigned to blocker $B_0$, $c_1$ to blocker $B_1$, $c_2$ and $c_3$ to detector $D_1$ and $c_4$ and $c_5$ to detector $D_0$.

Figure 11 $(a)$ depicts the case when none of the blockers are inserted and the result is composed of the bit string $c_5c_4c_3c_2$. The probability of detection in $D_1$ corresponds to the sum of the probabilities of 0001 and 0010, and the probability of detection in $D_1$, to the sum of the probabilities of 0100 and 1000, which are both around 50%. Fig. 11 $(b)$ corresponds to the case where $B_0$ is inserted in mode $D$ and the result is composed of the bit string $c_5c_4c_3c_2c_0$. The probability of a quanton being blocked by $B_0$ is approximately 50% (result 00001), and the probability of a detection in $D_0$ corresponds to the sum of the probabilities of 01000 and 10000, which is also around 50%. The case when blocker $B_1$ is inserted in mode L is illustrated in Fig. 11 $(c)$, in which the result is a bit string of the form $c_5c_4c_3c_2c_1$. The result 00001 corresponds to quantons blocked by $B_1$, which is the case where Afshar *et al.* perceived a reduction in photon count by the insertion of the wire grid. The bit strings corresponding to a detection in $D_0$ are the same for the previous case, and the probability of a detection in $D_1$ corresponds to the sum of the probabilities of 00100 and 00010. This case was constructed to simulate the amount of quantons (about 2%) that are blocked by the wire grid in the optical experiment. Finally, Fig. 11 $(d)$ depicts the case with both blockers in place, with the result being composed by the bit string $c_5c_4c_3c_2c_1c_0$. About 50% of the quantons are blocked by $B_0$ (result 000001) and around 1% by $B_1$ (result 000010). The probability of a detection in $D_0$ (48.5%) corresponds to the sum of the probabilities of 010000 and 100000, and the probability of a detection in $D_1$ (0.5%), to the probability of the result 000100.

The classical simulations closely adhere to the theoretical framework; however, similar to the modified Unruh's experiment, the demonstrative results are influenced by environmental interactions and noise in the quantum hardware. They reveal numerous instances where the photon count per mode is not preserved due to these imperfections. We omitted part of the demonstrative results where the photon count per mode is not preserved to improve visual clarity (the complete data can be found in the supplemental material [63]).

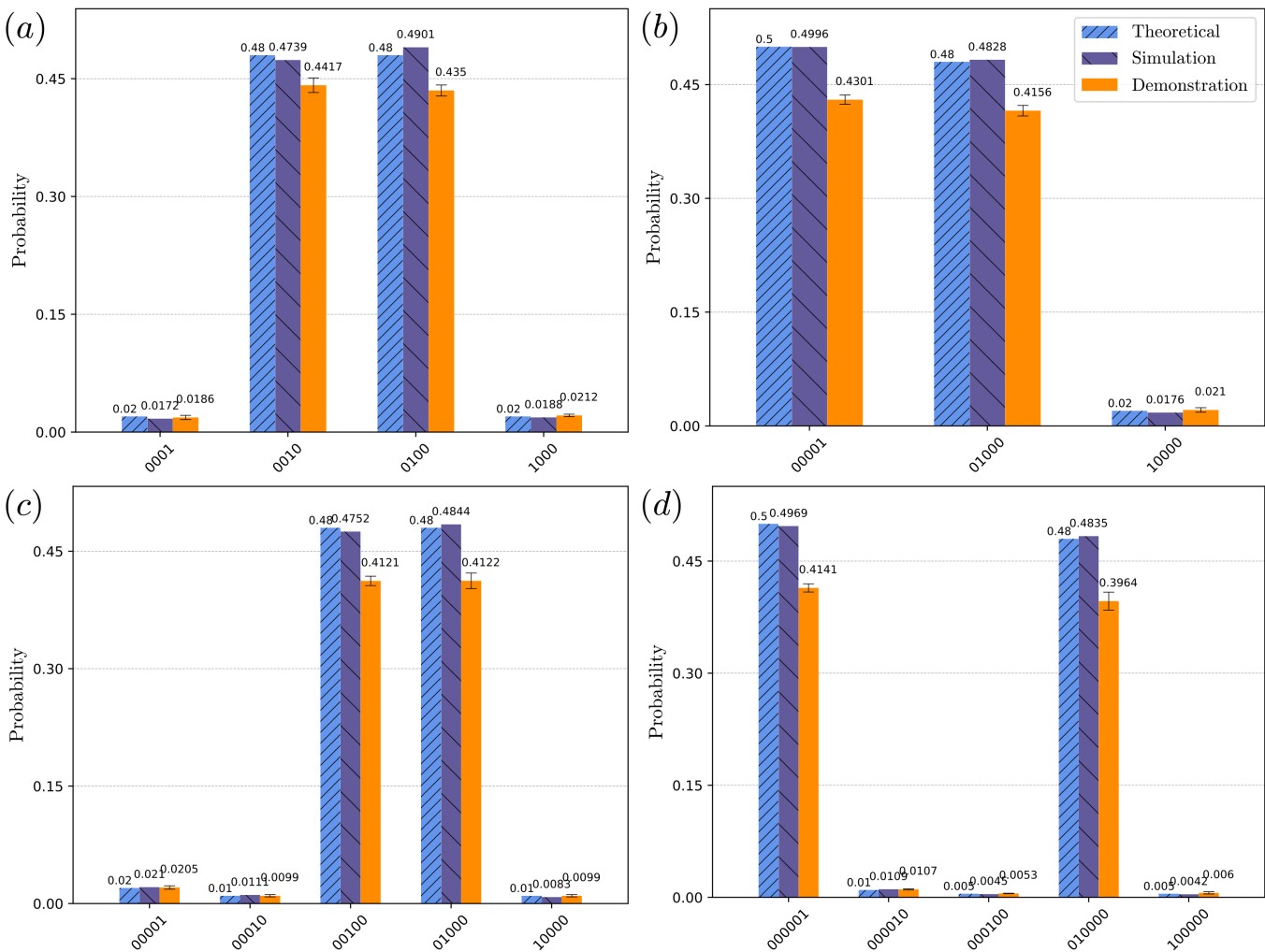

Figure 11: Histograms depicting the theoretical (bars with very close diagonal hatching), simulation (bars with widely spaced diagonal hatching), and demonstrations (solid bars) probability results for four different configurations of Pessoa Júnior's experiment circuit: ($a$) no blocker inserted and classical bits composed by $c_5c_4c_3c_2$. Detections in $D_1$ correspond to the sum of results 0001 and 0010, and detections in $D_0$ to the sum of results 0100 and 1000; ($b$) blocker $B_0$ inserted in mode D and classical bits composed by $c_5c_4c_3c_2c_0$. About 50% of photons hit the blocker (identified by the result 00001) and the other 50% pass through the upper arm (mode C) and are detected in $D_0$ (identified by the sum of results 01000 and 10000); ($c$) blocker $B_1$ inserted in mode L and classical bits composed by $c_5c_4c_3c_2c_1$. The 2% of quantons that hit blocker $B_1$ represent the photons that hit the wire grid in Afshar's experiment with both slits open. Detections in $D_0$ correspond to the sum of results 10000 and 01000, and detections in $D_1$ to the sum of results 00010 and 00100; and ($d$) both blockers inserted and classical bits composed by $c_5c_4c_3c_2c_1c_0$. The photons that are not blocked by $B_0$ (000001) nor $B_1$ (000010) are detected in both $D_1$ (000100) and $D_0$ (sum of results 010000 and 100000). The error bars correspond to the standard deviation for five repetitions of the quantum demonstrations, which were obtained with IBMQ's Heron r2 processor ibm_kingston quantum chip. The complete results (including those omitted to improve visual clarity) and the chip's calibration parameters can be found in the supplemental material [63].

## E.   Modified Unruh's experiment for the two-photon case

Now we shall discuss a new situation in the modified Unruh's experiment as depicted in Fig. 4. Instead of considering only one photon per mode, we consider a two-photon initial state $|\psi_0\rangle = |11\rangle_{\text{AB}}$, i.e., the mode B is populated as well. Appendix D provides a comprehensive overview of the evolved state progression after BS$_3$ and the blocker scenarios. This state, when passing through the BS$_1$, produces the well-known HOM interference state $|\psi_1\rangle = i\left(|20\rangle_{\text{CD}} + |02\rangle_{\text{CD}}\right)/\sqrt{2}$. The state after BS$_1$ involves a coherent superposition of only two Fock states—excluding the state $|11\rangle$ at the output. Given the similarity of this initial state with the scenario of a single-photon case after BS$_1$, if we insert a photon number-resolving detector in each path, it results in both photons being detected in only

one path per experiment. After the mirrors, phase $\phi_{\mathrm{E}}$ and $\mathrm{BS}_2$, the state is given by

$$|\psi_3\rangle = \frac{i\left(e^{i\phi_{\mathrm{E}}} - 1\right)}{2\sqrt{2}}\left(|20\rangle_{\mathrm{GH}} - |02\rangle_{\mathrm{GH}}\right) + \frac{1 + e^{i\phi_{\mathrm{E}}}}{2}|11\rangle_{\mathrm{GH}}. \tag{31}$$

After $\mathrm{BS}_2$, the state, which is dependent on $\phi_{\mathrm{E}}$, also does not represent the most extensive superposition achievable in a Hilbert space for two qubits. One way of quantifying the superposition between the states $|20\rangle$ and $|02\rangle$ is using a relation similar to the interferometric visibility. Usually, the visibility relation is defined [64] as $V_{1,1}(\phi_{\mathrm{E}}) = (C_{\max} - C_{\min})/(C_{\max} + C_{\min})$. Here, $C_{\max}$ and $C_{\min}$ are the maximal and minimal probabilities of obtaining coincidence (antibunching) clicks after $\mathrm{BS}_2$ in the detectors for the time of arrival of each photon before the $\mathrm{BS}_1$. So, in this case, there is a time distinguishability involved that we are not exploring in this work. In order to better reflect phase variation as an inference of superposition between states $|20\rangle$ and $|02\rangle$ and to perform a similar analysis to that done by Pessoa Júnior for the two-photon case, we define the bunching visibility as

$$V_{D_j}(\phi) = \frac{Pr(D_j)_{\max} - Pr(D_j)_{\min}}{Pr(D_j)_{\max} + Pr(D_j)_{\min}}, \quad j = 0, 1, \tag{32}$$

where in this context $Pr(D_j)_{\max}$ $(Pr(D_j)_{\min})$ is the maximum (minimum) probability, for a certain phase variation $\phi$, that both quantons will be registered in detector $D_j$ (bunching detection). In a conventional configuration of MZI, as presented in Fig. 2, Eq. (31) for the initial state $|\psi_0\rangle = |11\rangle_{\mathrm{AB}}$ reads $V_{D_j}(\phi_{\mathrm{E}}) = 1$. In other words, the bunching visibility is inferring the wave-like behavior of the superposition between the states $|20\rangle_{\mathrm{CD}}$ and $|02\rangle_{\mathrm{CD}}$. Therefore, from now on for the two-photon case, we will refer to the wave-like behavior as being the measure of this type of superposition through bunching visibility, unless otherwise stated.

To demonstrate the correspondence in the analysis made by Pessoa Júnior, blocking mode C [see Eq. (D8)] the quantons that pass through mode D lead us to obtain after $\mathrm{BS}_3$ the following normalized state

$$|\psi_5'\rangle = -\frac{i}{4}\left\{\left(1 + 3e^{i\phi_{\mathrm{H}}}\right)|20\rangle_{\mathrm{KL}} - \left(1 - e^{i\phi_{\mathrm{H}}}\right)\left[|02\rangle_{\mathrm{KL}} + i\sqrt{2}\,|11\rangle_{\mathrm{KL}}\right]\right\}. \tag{33}$$

Setting $\phi_{\mathrm{H}} = 0$, results in only detector $D_0$ registering clicks. By blocking mode D, detector $D_1$ is the only one that registers clicks, as can be noted in Eq. (D11). So, the arm-detector relation is established in the analysis provided above in a very similar way to that in the modified Unruh's experiment.

Now, with mode C and D free, similarly to the extrapolation made by Pessoa Júnior for interferometric visibility, we will extrapolate the use of bunching visibility $V_{D_0}$ after $\mathrm{BS}_3$ to infer the quanton's behavior after $\mathrm{BS}_1$ and after $\mathrm{BS}_2$. To analyze the behavior in $\mathrm{MZI}_1$, we need to set up $\phi_{\mathrm{H}} = 0$ to ensure the path-detector relationship as noted before. The state in Eq. (D5) with this phase is reduced to

$$|\psi_5(\phi_{\mathrm{E}}, \phi_{\mathrm{H}} = 0)\rangle = -\frac{i}{\sqrt{2}}\left(|20\rangle_{\mathrm{KL}} + e^{i\phi_{\mathrm{E}}}|02\rangle_{\mathrm{KL}}\right). \tag{34}$$

The variation of $\phi_{\mathrm{E}}$ leads us to conclude that $V_{D_0}(\phi_{\mathrm{E}}) = 0$ by the bunching visibility in Eq. (32) and the respective associated behavior in $\mathrm{MZI}_1$ is particle-like as we are considering a pure state. Now, in the $\mathrm{MZI}_2$ the desired behavior is a superposition between the states $|20\rangle$ and $|02\rangle$; for this we set up $\phi_{\mathrm{E}} = \pi$ and the state in Eq. (D5) is reduced to

$$|\psi_5(\phi_{\mathrm{E}} = \pi, \phi_{\mathrm{H}})\rangle = -\frac{i\left(1 + e^{i\phi_{\mathrm{H}}}\right)}{2\sqrt{2}}\left(|20\rangle_{\mathrm{KL}} + |02\rangle_{\mathrm{KL}}\right) - \frac{\left(1 - e^{i\phi_{\mathrm{H}}}\right)}{2}|11\rangle_{\mathrm{KL}}. \tag{35}$$

The variation of $\phi_{\mathrm{H}}$ results in $V_{D_0}(\phi_{\mathrm{H}}) = 1$ by the bunching visibility in Eq. (32), characterizing a wave-like behavior as expected. The graph of probabilities by phase variation is presented in Fig. 13. The results are extracted from the circuit presented in Fig. 12.

In contrast with the circuits illustrated in Figs. 6 and 10, at least two qubits per mode are needed to simulate two photons (a base-2 system requires more than one digit to represent the number 2). Here, qubits $q_0$ and $q_1$ correspond to mode $A$, while $q_2$ and $q_3$ correspond to mode $B$. The classical bits $c_0$, $c_1$ represent detector $D_0$, and $c_2$, $c_3$ detector $D_1$. The circuit begins in the state $|\psi_0\rangle = |0101\rangle \mapsto |11\rangle_{\mathrm{AB}}$ and evolves according to beam splitter and mirror unitary operations. The outcomes labeled as 1100, 0011, and 0101 in Fig. 13 correspond to "both $D_0$", "both $D_1$", and "coincidence", respectively.

This case is notably similar to the analysis conducted for the modified Unruh's experiment, and the same conclusions can be extracted as before. It is important to highlight that there are apparent limitations for the two-photon case as there are for just one photon. For the two experiments, the limitation is on the order of the initial state and also for a rigid experimental configuration, for example, not varying $(T, R)$. Particularly, for the two-photon case, the similarity with one-photon case is only achieved with the initial state $|11\rangle_{\mathrm{AB}}$, in a more general initial state, the relation path-detector is broken. In the next section, we will explore the QCP view of the two-photon case and highlight that these limitations do not restrict the evaluation of wave-particle duality through this framework.

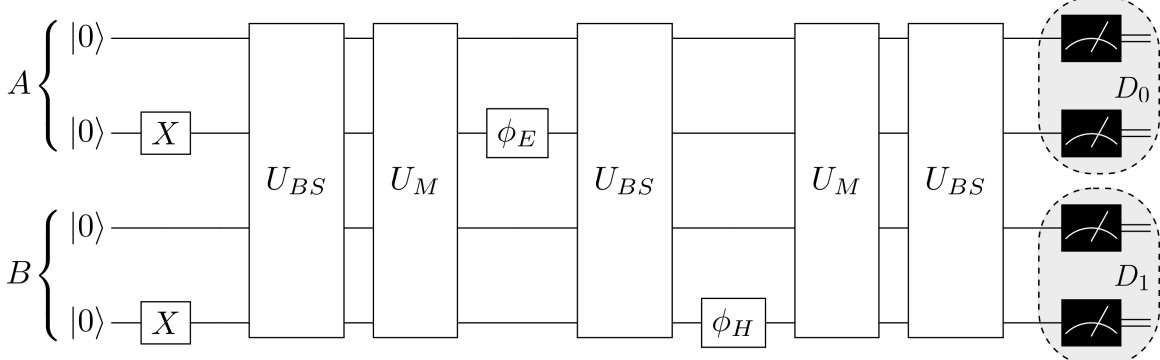

Figure 12: Quantum circuit used in the simulations of Fig. 13. The difference from the circuits of Figs. 6 and 10 is that at least two qubits are necessary to represent each mode, since we are interested in simulating two photons (one digit is not enough to represent the number 2 in a base-2 encoding). Qubits $q_0$ and $q_1$ represent mode $A$, $q_2$ and $q_3$ mode $B$, the classical bits $c_0$ and $c_1$ represent the detector $D_0$ and $c_2$ and $c_3$ detector $D_1$. The circuit is prepared in the state $|\psi_0\rangle = |0101\rangle \mapsto |1_A, 1_B\rangle$, which is prepared with the $X$ gate in each mode before the first BS, and evolved according to the beam splitter and mirror unitary operators as before. The possible measurement results are 1100 (bunching detection in $D_0$), 0011 (bunching detection in $D_1$) and 0101 (antibunching detection), which are labeled in Fig. 13 as "Both $D_0$", "Both $D_1$" and "Coincidence", respectively.

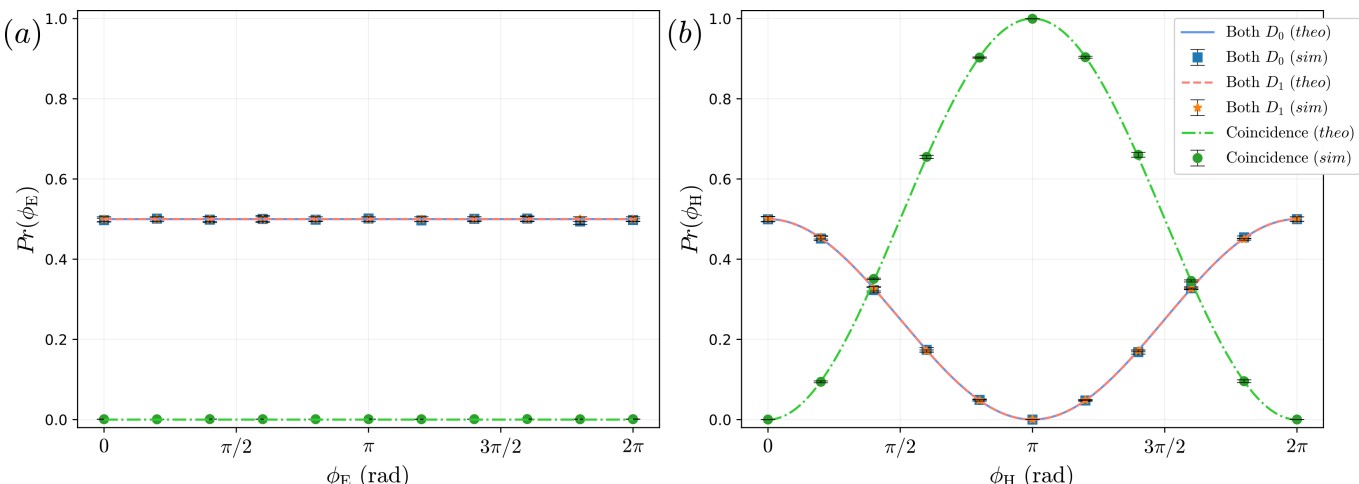

Figure 13: Detection probabilities as a function of phase variation for the quantum circuit corresponding to the modified Unruh's experiment with two photons (Fig. 12). In the main text it was identified that the phases need to be $\phi_E = \pi$ (for variations of $\phi_H$) and $\phi_H = 0$ (for variations of $\phi_E$). The phase $\phi_H = 0$ is necessary to guarantee that both photons passing through one of the arms are related to one specific detector. (a) By setting $\phi_H = 0$ and varying $\phi_E$, it is possible to infer that there is no wave-like behavior ($V_{D_0} = 0$), as stated by Pessoa Júnior, since the probabilities of the possible measurement outcomes (both photons detected in $D_0$, labeled "both $D_0$" and both detected in $D_1$, labeled "both $D_1$") are phase independent. (b) On the other hand, by setting $\phi_E = \pi$ and varying $\phi_H$, we can infer that there is a wave-like behavior represented by the superposition between the states $|20\rangle$ and $|02\rangle$ ($V_{D_0} = 1$). In both plots the lines represent the theoretical probabilities and the points were obtained by classical simulation of the quantum circuits; the error bars correspond to the standard deviation for five repetitions of these simulations.

### F. Discussion and analysis of the experiments in the light of the quantum complementarity principle

In the context of complementarity, experiments utilizing interferometry typically seek to elucidate the dual characteristics of the quanton after interacting with the first beam splitter. This is achieved by creating complementarity relations tailored specifically for the whole experimental arrangement. Significant efforts have been made to modify these sets of controls and develop alternative approaches to construct complementarity relations. However, despite these approaches, the fundamental objective remains the same as in the MZI depicted in Fig. 2: determining the quanton's behavior after the first BS.

A key distinction in the modified version of Unruh's experiment lies in the functions used to quantify the particle-like

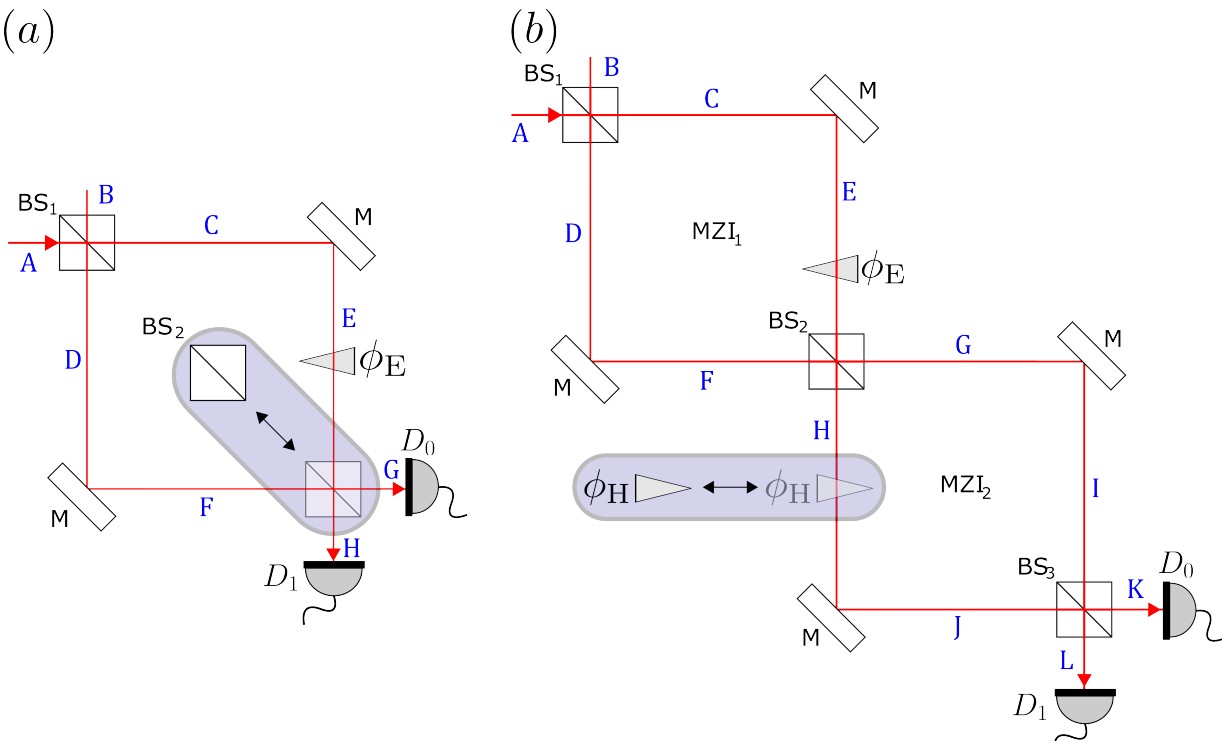

Figure 14: Interferometric experiments manifest a counterintuitive concept regarding causality through the delayed-choice paradigm in Quantum Mechanics. (*a*) Wheeler's delayed-choice experiment. There are two configurations in consideration: 1) with $BS_2$ the definition is that there is a particle-like behavior, since the click reveals the path taken by the photon and also the probabilities are not phase sensitive (no interference observed). 2) With $BS_2$ in place the interference is observed by phase variation. Postponing the decision to include or not $BS_2$ in the photon path, after the photon passes through $BS_1$, according to Wheeler, it is possible to change the wave-particle properties of the quanton. (*b*) We follow the same reasoning used by Pessoa Júnior in the modified Unruh's experiment to construct a delayed-choice in this experimental setup. By setting $\phi_E = \pi/2$ and $\phi_H = 0$ (the same as not including the optical device), Pessoa Júnior argues that there is a particle-like behavior in $MZI_1$ and wave-like behavior in $MZI_2$, reflecting each step in Afshar's experiment. Keeping $\phi_E = \pi/2$ and by setting $\phi_H = \phi/2$ both MZI manifest wave-like behavior. If we postpone the choice of whether or not to include $\phi_H$, after the photon has already left $MZI_1$, we can change the behavior of the wave-particle duality in $MZI_1$.

($P$) and wave-like ($W$) behaviors. Many complementarity relations—particularly those that rely on IV to quantify $W$—invoke retroactive inference, as quantification does not occur in the region where wave-like behavior manifests itself. Afshar, however, shifts this perspective by using retroactive inference to quantify $P$, while $W$ is inferred non-destructively via the wire grid precisely where the phenomenon manifests.

Pessoa Júnior, in his analysis, argued that Afshar's experiment does not violate the CR. Instead, by shifting the viewpoint from Bohr's interpretation of the experiment as a whole to an analysis at each stage of the process, he concluded that complementarity remains valid at every step. This perspective aligns closely with QCP, but by adopting an alternative formalism that doesn't rely on retroactive inference and also with different conclusions. To explore in more depth the ideas behind the approach taken by Pessoa Júnior, we will discuss more about Wheeler's delayed-choice experiment. Next, we will present a new variation of a delayed-choice experiment that arises in the modified Unruh's experiment. We will engage in a discussion regarding the retro-inference issue, the absence of a formal definition for Bohr's complementarity principle, and a renewed perspective in terms of the QCP. A further novelty is investigated by examining the modified Unruh's experiment applied to the two-photon case.

### 1. More about delayed-choice experiments

To elaborate our evaluation behind the approach of Pessoa Júnior, we need to recover Wheeler's delayed-choice experiment (WDCE) as presented in Fig. 14 (a) and the definition of IV in Eq. (18). According to WDCE, after the quanton passes through $BS_1$, one can choose whether or not to insert $BS_2$, and by delaying the choice—after the quanton traverses $BS_1$—one can change by retrodiction the way the quanton will behave: particle-like behavior in the

absence of $BS_2$ or wave-like behavior with $BS_2$ in place.

In the literature, the definition of particle-like behavior in the absence of $BS_2$ is motivated by a click on the detector that reveals the path taken by the quanton, despite the fact that the state of the quanton, by Schrödinger's equation, is described by a quantum superposition. More about this question and a new way to quantify the superposition after the $BS_1$ through entanglement is proposed and discussed in Ref. [65]. Eq. (18) also allows us to confirm through IV that the probabilities remain unchanged as we vary the phase, indicating consistency. On the other hand, the inclusion of $BS_2$ makes the probabilities sensitive to phase variation, indicating a wave-like behavior. This kind of experiment is a conventional example in the literature for the first version of BCP with complementary experimental setup: $P = 1$ and $W = 0$ without the $BS_2$, and $P = 0$ and $W = 1$ when $BS_2$ is inserted.

It should be emphasized that the conceptualization of the delayed-choice experiment with MZI (around 1977) is constructed before the first updated BCP in quantitative terms constructed by Greenberger-Yasin in 1988, when partial quantifications appear. Analyzing the case where $T \neq R$ in WDCE and using the predictability in Eq. (19) to quantify particle-like behavior leads to the following scenario: 1) Without $BS_2$, $\mathcal{V} = 0$ and $\mathcal{P} = |T_1^2 - R_1^2|$, i.e., it depends on the experimental configuration to determine $(T_1, R_1)$; 2) With $BS_2$, $\mathcal{V} = 2TR$, but the predictability still reads $\mathcal{P} = |T_1^2 - R_1^2|$. In scenario 1), we have $W + P = 1$ because the quantum system is pure and there is only a single degree of freedom under consideration. Therefore, we expect that $W$, $P$, or the sum of both gives us equality. Due to $W$ being null as a consequence that the probabilities remain the same despite phase changes, it leads us to conclude that $P$ must be equal to 1. However, this condition is not satisfied when $(T, R) \neq \{(0, 1), (1, 0)\}$, leading to a violation of the equality. Although limited, let's consider that WDCE is only valid with balanced $BS_1$. Therefore, the logic of using interferometric visibility without $BS_2$ appears to be feasible, but this situation also makes the extrapolation made by Pessoa Júnior viable. Furthermore, Greenberger-Yasin CR does not impose restrictions on the use of $\mathcal{V}$. We will see below that this lack of formal definition of the complementarity principle allows for another variation of delayed-choice in the modified version of the Unruh's experiment.

In the discussion that comes next we are no longer elaborating on the variation of $(T, R)$ in Fig. 14 (b), as mainly adopted in Ref. [47], because the same reasoning can be constructed. Our main goal is to delve into further details on the method employed by Pessoa Júnior in expanding the use of IV to describe wave-like behavior, and a novel variation of the delayed-choice experiment observed in the modified Unruh's experiment.

In the analysis constructed in Sec. II C, the choice of $\phi_E = \pi/2$ and $\phi_H = 0$ allows one to relate the particle-like behavior in $MZI_1$ and the wave-like behavior in $MZI_2$. For instance, if we analyze the configuration using the Greenberger-Yasin CR, $P = 0$ is independent of the variation $\phi_E$.

In his work, Pessoa Júnior also mentions another case where $\phi_E = \phi_H = \pi/2$. Fixing one phase and varying the other, following the same reasoning as before, one can conclude that in both MZIs the behavior is wave-like, i.e., fixing $\phi_E = \pi/2$ or $\phi_H = \pi/2$ and varying the other, the probabilities are phase-sensitive.

In summary, the set $\phi_E = \pi/2$ and $\phi_H = 0$ is related to particle-like and wave-like in $MZI_1$ and $MZI_2$, respectively; while the set $\phi_E = \pi/2$ or $\phi_H = \pi/2$ produces wave-like behavior in both MZIs. Now comes the new variation of the delayed-choice experiment. By maintaining the $\phi_E = \pi/2$ in place, if we postpone the choice of including or not $\phi_H = \pi/2$ [as represented in Fig. 14 (b)] after the quanton passes through to $BS_2$ led us to construct a new variation of the delayed-choice experiment that changes the behavior of the quanton in $MZI_1$ even after the quanton leaves $MZI_1$. This is feasible because, according to Bohr, the experiment must be considered in its entirety, a phase sensitivity in the configuration through IV leads us to conclude that the behavior can be changed by choosing to include or not the phase $\phi_H = \pi/2$. Although we do not analyze this experimental setup in Ref. [47], the results obtained there can be perfectly compatible with this new experimental version. The conclusions will be exactly the same from there, that is, it is not possible to modify the quanton's behavior retroactively. Certainly, here we could explore much more general cases in terms of states and the variability of $(T, R)$ in which the QCP would be consistent without invoking retroactivity or relying on delayed-choice effects.

### 2. QCP in the modified Unruh's experiment for the two-photon case

As the analysis through the QCP for the new variation of the delayed-choice experiment discussed above can be extracted from the results obtained in Ref. [47], here we are focused on obtaining and discussing the case with two-photons which has not been explicitly explored before in the context of QCP. In the modified Unruh's experiment for the two-photon case, the access to the whole Hilbert space of two qubits is limited by the nature of bosonic systems, i.e., there is no access to states as $|00\rangle$, $|10\rangle$ and so on, which do not preserve the quantity of two photons in the quantum state. To proceed with the analysis of QCP, we can identify that this case can be simplified by an effective three-level quantum system. We will define the states obtained in Appendix D as the following: $|20\rangle \equiv |0\rangle$, $|02\rangle \equiv |1\rangle$, and $|11\rangle \equiv |2\rangle$. The CR with $d = 3$ and noting that the quantum system is pure, the CR presented in Eq. (16)

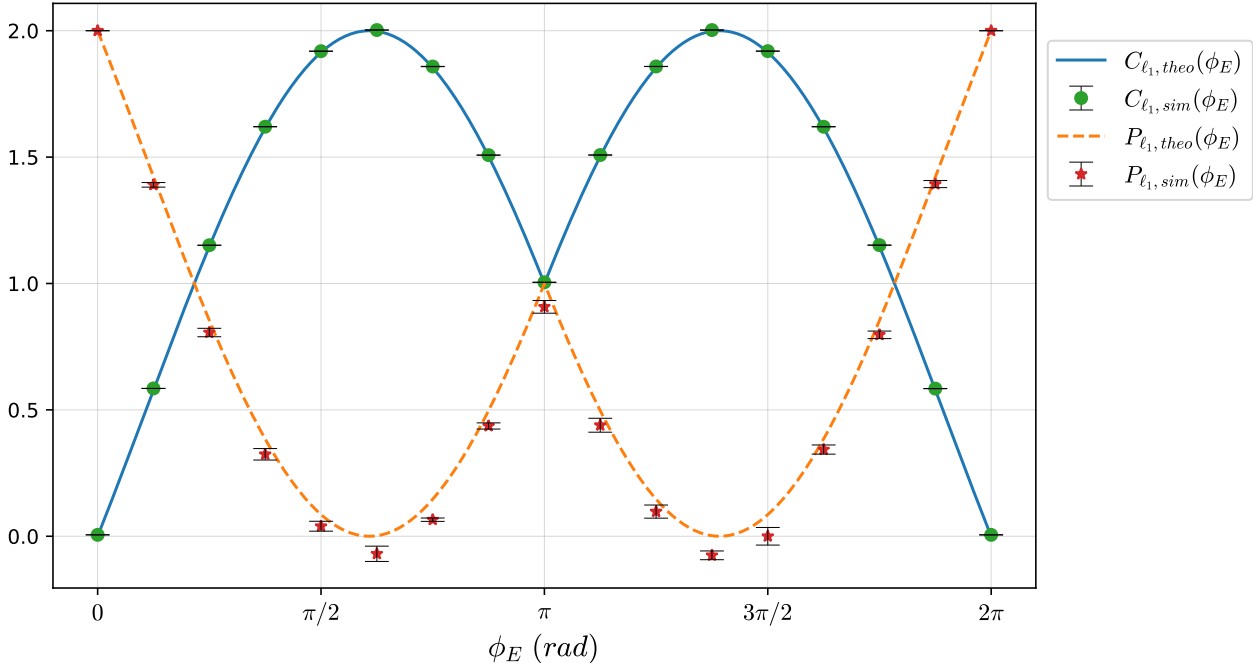

Figure 15: Interplay between the functions in the quantum complementarity relations obtained in Eqs. (38) and (39) (for the two-photon Unruh's experiment after the second beam splitter). The lines correspond to the theoretical values and the points to the classical simulation results. The density matrix of the system for each $\phi_E$ was constructed via quantum state tomography implemented with Qiskit's EstimatorV2 primitive, in which were used $2^{23}$ shots to estimate the average values and 100 Trotter steps to construct the corresponding circuit. The error bars were obtained by repeating this procedure five times for each point and then calculating the standard deviation of the results. The realization of this process on current quantum hardware is unfeasible given the depth of the circuit used.

becomes

$$C_{l_1}(\rho_t) + P_{l_1}(\rho_t) = 2. \tag{36}$$

For example, the state in Eq. (D1) leads us to $C_{l_1}(\rho_1) = P_{l_1}(\rho_1) = 1$, which is reasonable as this pure qubit state has unbalanced weights between the available states in the effective Hilbert space. The state $|11\rangle$ does not appear in the state, suggesting an increase in the path information available; employing the analogy of a game, no one would bet on the photons taking separate paths. Although the uncertainty is maximum between knowing which state will emerge between $|20\rangle$ and $|02\rangle$, there is a certain amount of information available about the state $|11\rangle$. As in a two-level system, predictability increases when the probability of a quanton passing through one of the two arms of the MZI increases, for example, by changing $(T, R)$ in the first BS or with a different initial state other than $|10\rangle_{AB}$. In the case of two bosons, note that there are three configurations available: both quantons passing through the upper arms, both passing by the lower arm, or each of them being detected in different arms of the MZI. The predictability is available by acknowledging that both quantons will not be registered in different arms of the MZI for this configuration; on the other hand, there is a quantity of wave-like behavior represented by the superposition between the state $|20\rangle$ and $|02\rangle$.

The more compelling case is presented after the BS$_2$. Equation (D3) is given by

$$|\psi_3\rangle = \frac{i\left(e^{i\phi_E} - 1\right)}{2\sqrt{2}}\left(|0\rangle_{\text{GH}} - |1\rangle_{\text{GH}}\right) + \frac{1 + e^{i\phi_E}}{2}|2\rangle_{\text{GH}}. \tag{37}$$

The functions in the CR for the wave-particle duality are obtained as

$$C_{l1}(\rho_3) = \sqrt{2}|\sin(\phi_E)| - \frac{\cos(\phi_E)}{2} + \frac{1}{2}, \tag{38}$$

$$P_{l1}(\rho_3) = -\sqrt{2}|\sin(\phi_E)| + \frac{\cos(\phi_E)}{2} + \frac{3}{2}, \tag{39}$$

which are phase dependent, because the choice of $\phi_E$ can produce a variety of states after $BS_2$. The interplay between the CR for a variation of $\phi_E$ can be found in Fig. 15. When $\phi_E$ assumes the values 0 or $2\pi$ the state in Eq. (37) is reduced to $|\psi_3\rangle = |11\rangle$ and there is maximum predictability, i.e., we know for sure that each quanton goes through different arms. For $\phi_E = \pi$, the analysis is the same for the state of Eq. (D1) and the result is $C_{l_1}(\rho_3) = P_{l_1}(\rho_3) = 1$. When $\phi_E$ assumes the values $2\arcsin\sqrt{2/3}$ or $2\pi - 2\arcsin\sqrt{2/3}$, there is the maximum manifestation of wave-like behavior, $C_{l_1}(\rho_3) = 2$, i.e., the coefficients for each state are balanced.

The analysis of the functions of quantum coherence and predictability that form a CR presented in Fig. 15 highlights the richness of wave-like and particle-like behaviors in the modified Unruh's experiment with the two-photon case as well. The dependency of $\phi_E$ in Eqs. (39) after the $BS_2$ shows that a variety of scenarios are constructed in this region for the wave-particle duality. Furthermore, although subsequent steps of the experiment do not bring novelty to the study, the action of $BS_3$ results in a collective dependence of the CR on parameters $\phi_E$ and $\phi_H$ as follows

$$
\begin{aligned}
C_{l_1}(\rho_5) &= \frac{(s_1 + s_2)\sqrt{1 - \cos\phi_E}\sqrt{1 - \cos\phi_H}}{4} + \frac{s_1 s_2}{8}, \\
P_{l_1}(\rho_5) &= 2 - \frac{(s_1 + s_2)\sqrt{1 - \cos\phi_E}\sqrt{1 - \cos\phi_H}}{4} - \frac{s_1 s_2}{8},
\end{aligned}
\tag{40}
$$

where

$$
\begin{aligned}
s_1 &= \sqrt{2\cos\phi_E + 2\cos\phi_H - 3\cos(\phi_E - \phi_H) + \cos(\phi_E + \phi_H) + 6}, \\
s_2 &= \sqrt{2\cos\phi_E + 2\cos\phi_H + \cos(\phi_E - \phi_H) - 3\cos(\phi_E + \phi_H) + 6},
\end{aligned}
$$

which is an additional exemplification of the versatility of QCP. Additionally, the initial state constraint is not essential for a QCP perspective analysis. It suffices to determine the effective Hilbert space for evaluating the experiment, thus choosing an initial state different from $|11\rangle_{AB}$ does not add complexity, as demonstrated in the analysis following the second and third beam splitters.

## III.   FINAL REMARKS

In this work, we explored the digital quantum simulation of interferometric alternatives to Afshar's experiment. Using the formalism of Refs. [9–11], which map bosonic operators to Pauli operators, we simulated two versions of nested MZIs proposed by Pessoa Júnior. Although second quantization is unnecessary in both the modified Unruh's and Pessoa Júnior's experiments without blockers, introducing blockers adds complexity. To address this, we utilize the previously mentioned formalism to effectively model the "absorption" by the blockers. In the absence of absorption, the experiment remains unchanged, enabling us to express both cases within a single quantum circuit. This approach allows a clearer examination of the role of blockers using quantum computer simulations. The extension for the two-photon case in the modified Unruh's experiment results in a similar analysis to that of the one-photon case.

By providing a digital quantum simulation of the nested MZI, this work also examines the issues raised by Afshar regarding the potential violation of the BCP. Afshar's approach, which relies on non-destructive measurements to quantify the wave-like behavior of the quanton, combined with the lack of a formal definition of the BCP, allowed for the construction of an *ad hoc* CR that seemingly violates the BCP.

Our digital quantum simulation offers both a classical simulation and demonstrative results on quantum computers, which show strong agreement with theoretical predictions. Pessoa Júnior's analysis of the modified Unruh's experiment incorporates a phase into each MZI, concluding that each stage of the experiment corresponds to specific behaviors analogous to those observed in Afshar's original experiment. Inspired by these findings, Pessoa Júnior proposed a new experimental setup to simulate additional aspects of Afshar's experiment. Furthermore, our simulations reveal that the observed 2% reduction in photon count caused by the wire grid can be replicated when both paths are free and with $B_1$ in place. The inclusion of the blocker $B_0$ in mode D, alongside $B_1$, successfully simulates detection at the opposing detector, consistent with the experimental results of Afshar. However, it should be noted that this version of the interferometric setup still cannot fully replicate all the nuances explored in the original experiment.

We performed a study on the modified Unruh's experiment for the two-photon case. By setting an initial state containing one photon in each mode, we constructed a very similar case to the original one presented by Pessoa Júnior. We defined a phase-sensitive visibility function to infer the wave-like behavior (consistent with the well-known HOM superposition), which enables us to draw the similar conclusions as for the single photon case. The restriction is that we only consider the initial state as $|11\rangle_{AB}$.

We also further explored the question of retro-inference in a new version of the delayed-choice experiment. By considering the phase variation approach to infer the quanton wave-like behavior for different regions of nested MZI,

we can implement a delayed-choice experiment that can modify the wave-like behavior in $\mathrm{MZI}_1$ even after the photon has already left this interferometer. The QCP provides a different perspective for both the original delayed-choice experiment discussed by Wheeler and for this new variation. The interpretation of delayed-choice setups in terms of the updated QCP does not lead to retro-causality in the quanton's behavior, and results in a different notion of wave-particle duality in the analysis of these experiments.

For the two-photon case, as expected, the QCP analysis gives us a different perspective. The collective dependence of the phase parameters in the final stages of the experiment highlights the potential of the QCP for broader investigations of wave-particle duality in many-body systems. From the point of view of the modified Unruh experiment as an analogy for the Afshar experiment, the choice of the initial state as being $|11\rangle_{\mathrm{AB}}$ is crucial for us to be able to relate the detectors to the path taken by the quantons. On the other hand, this initial state restriction is not necessary to analyze the experiment from a QCP perspective. All that is needed is to identify the effective Hilbert space to evaluate the experiment, and therefore, an initial state other than $|11\rangle_{\mathrm{AB}}$ would not introduce additional complications, as we saw in the analysis after the second and third beam splitters. Future work could involve more general initial states including more quantons, alternative experimental setups, and also exploring the interface between the quantum and classical domains.

By revisiting how the QCP solves paradoxes, such as those posed by Afshar's experiment, this work also demonstrates that quantum computers are highly adaptable and effective platforms for simulating experiments involving bosonic systems. These findings further reinforce the utility of digital quantum simulations in expanding the investigations on QM fundamentals, in particular those involving configurations that deviate from conventional ones, such as the modified Unruh's experiment with two photons.

### Acknowledgments

This work was supported by the Coordination for the Improvement of Higher Education Personnel (CAPES) under Grant No. 88887.827989/2023-00, by the Research Support Foundation of the State of Rio Grande do Sul (FAPERGS) under Grant No. 23/2551-0001199-9, by the National Council for Scientific and Technological Development (CNPq) under Grants No. 309862/2021-3, No. 409673/2022-6, No. 421792/2022-1, and No. 132266/2025-3, and by the National Institute for the Science and Technology of Quantum Information (INCT-IQ) under Grant No. 465469/2014-0.

### Appendix A: Relations between input and output bosonic operators for arbitrary $\theta$

For illustration, let us examine the relationship between input modes A and B and output mode C. In this case, we have the following

$$c = U_{\mathrm{BS}}^{\dagger}\, a\, U_{\mathrm{BS}} = \left(e^{i\theta(b^{\dagger}a + ba^{\dagger})}\right)^{\dagger} a\, e^{i\theta(b^{\dagger}a + ba^{\dagger})} = e^{-i\theta(ab^{\dagger} + a^{\dagger}b)} a\, e^{i\theta(b^{\dagger}a + ba^{\dagger})} = e^{-i\theta(b^{\dagger}a + ba^{\dagger})} a\, e^{i\theta(b^{\dagger}a + ba^{\dagger})}, \qquad (A1)$$

since $[a, b^{\dagger}] = 0$ and $[a^{\dagger}, b] = 0$. Using the Baker−Hausdorff lemma,

$$e^{\gamma H} a\, e^{-\gamma H} = a + \gamma[H, a] + \frac{\gamma^2}{2!}\big[H, [H, a]\big] + \frac{\gamma^3}{3!}\big[H, [H, [H, a]]\big] + \cdots, \qquad (A2)$$

with $\gamma = -i\theta$ and $H = b^{\dagger}a + ba^{\dagger}$, it follows that

$$c = a - i\theta\big[b^{\dagger}a + ba^{\dagger}, a\big] - \frac{\theta^2}{2!}\big[b^{\dagger}a + ba^{\dagger}, [b^{\dagger}a + ba^{\dagger}, a]\big] + \cdots. \qquad (A3)$$

Given that

$$[b^{\dagger}a + ba^{\dagger}, a] = [b^{\dagger}a, a] + [ba^{\dagger}, a] = b^{\dagger}[a, a] + [b^{\dagger}, a]\, a + b[a^{\dagger}, a] + [b, a]\, a^{\dagger} = -b[a, a^{\dagger}] = -b, \qquad (A4)$$

and

$$\begin{aligned}
[b^{\dagger}a + ba^{\dagger}, [b^{\dagger}a + ba^{\dagger}, a]] &= [b^{\dagger}a + ba^{\dagger}, -b] = -\big([b^{\dagger}a, b] + [ba^{\dagger}, b]\big) \\
&= -\big(b^{\dagger}[a, b] + [b^{\dagger}, b]\, a + b[a^{\dagger}, b] + [b, b]\, a^{\dagger}\big) = -\big(-[b, b^{\dagger}]\, a\big) = a,
\end{aligned} \qquad (A5)$$

one obtains

$$\begin{aligned}
c &= a + i\theta b - \frac{\theta^2}{2!}a - i\frac{\theta^3}{3!}b + \frac{\theta^4}{4!}a + i\frac{\theta^5}{5!}b + \cdots = a\left(1 - \frac{\theta^2}{2!} + \frac{\theta^4}{4!} + \cdots\right) + ib\left(\theta - \frac{\theta^3}{3!} + \frac{\theta^5}{5!} + \cdots\right) \\
&= a\cos(\theta) + ib\sin(\theta),
\end{aligned} \qquad (A6)$$

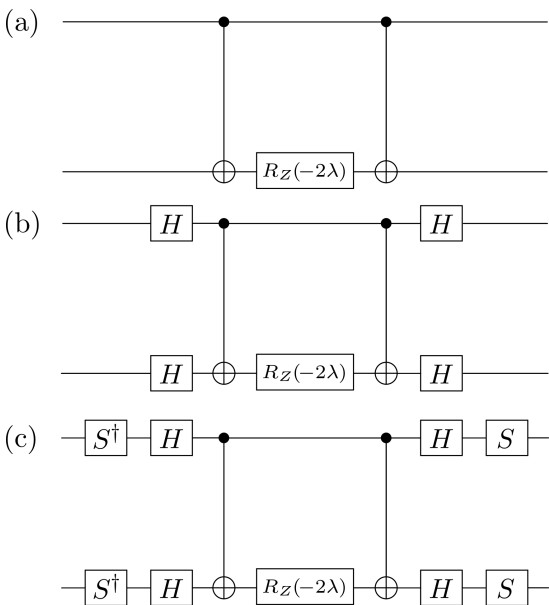

Figure 16: Quantum circuits that implement the unitary transformations (a) $e^{i\lambda(Z\otimes Z)}$, (b) $e^{i\lambda(X\otimes X)}$ and (c) $e^{i\lambda(Y\otimes Y)}$. In the last circuit, the gates $S^\dagger H$ could be replaced by a single $R_X(\pi/2)$, and $HS$ by $R_X(-\pi/2)$.

and thus $c^\dagger = a^\dagger \cos(\theta) - ib^\dagger \sin(\theta)$. Similarly, this holds for the relationship between input modes A and B and the output mode D:

$$d = ia\sin(\theta) + b\cos(\theta), \quad d^\dagger = -ia^\dagger \sin(\theta) + b^\dagger \cos(\theta). \tag{A7}$$

Eqs. (2) and (3) are obtained by defining $T = \cos(\theta)$ and $R = \sin(\theta)$.

## Appendix B: Beam splitter unitary implementation

In order to implement the unitary transformation of the BS, it is initially useful to implement the unitary $U = \exp\left[i\lambda(Z\otimes Z)\right]$. The spectral decomposition of $Z\otimes Z$ is given by

$$Z\otimes Z = \left(|0\rangle\langle0| - |1\rangle\langle1|\right)\otimes\left(|0\rangle\langle0| - |1\rangle\langle1|\right) = \left(|00\rangle\langle00| + |11\rangle\langle11|\right) - \left(|01\rangle\langle01| + |10\rangle\langle10|\right), \tag{B1}$$

and consequently, the spectral decomposition of $U$ reads

$$\exp\left[i\lambda(Z\otimes Z)\right] = e^{i\lambda}\left(|00\rangle\langle00| + |11\rangle\langle11|\right) + e^{-i\lambda}\left(|01\rangle\langle01| + |10\rangle\langle10|\right). \tag{B2}$$

Noting that

$$C_X^{0\to1}\left(\mathbb{I}\otimes e^{i\lambda Z}\right)C_X^{0\to1} = e^{i\lambda}\left(|00\rangle\langle00| + |11\rangle\langle11|\right) + e^{-i\lambda}\left(|01\rangle\langle01| + |10\rangle\langle10|\right) = \exp\left[i\lambda(Z\otimes Z)\right] \tag{B3}$$

and that

$$e^{i\lambda Z} = e^{i\lambda}|0\rangle\langle0| + e^{-i\lambda}|1\rangle\langle1| = e^{i\lambda}\begin{bmatrix}1 & 0 \\ 0 & 0\end{bmatrix} + e^{-i\lambda}\begin{bmatrix}0 & 0 \\ 0 & 1\end{bmatrix} = \begin{bmatrix}e^{i\lambda} & 0 \\ 0 & e^{-i\lambda}\end{bmatrix} = R_Z(-2\lambda), \tag{B4}$$

the unitary $U$ can be written as

$$\exp\left[i\lambda(Z\otimes Z)\right] = C_X^{0\to1}\left[\mathbb{I}\otimes R_Z(-2\lambda)\right]C_X^{0\to1}, \tag{B5}$$

which is implemented in a quantum circuit as shown in Fig. 16 (a).

To implement the terms of $U_{\mathrm{BS}}$, we begin by writing their spectral decomposition as

$$\exp\left[i\lambda(X\otimes X)\right] = e^{i\lambda}\left(|++\rangle\langle++| + |--\rangle\langle--|\right) + e^{-i\lambda}\left(|+-\rangle\langle+-| + |-+\rangle\langle-+|\right) \tag{B6}$$

and

$$\exp\left[i\lambda(Y \otimes Y)\right] = e^{i\lambda}\left(|\oplus\oplus\rangle\langle\oplus\oplus| + |\ominus\ominus\rangle\langle\ominus\ominus|\right) + e^{-i\lambda}\left(|\oplus\ominus\rangle\langle\oplus\ominus| + |\ominus\oplus\rangle\langle\ominus\oplus|\right), \tag{B7}$$

where $|\pm\rangle = (|0\rangle \pm |1\rangle)/\sqrt{2}$, $|\oplus\rangle = (|0\rangle + i|1\rangle)/\sqrt{2}$, and $|\ominus\rangle = (|0\rangle + i|1\rangle)/\sqrt{2}$. Since the Hadamard gate $H$ switches between the computational basis and the $X$ basis, one can write

$$\exp\left[i\lambda(X \otimes X)\right] = H \otimes H \left[e^{i\lambda}(|00\rangle\langle00| + |11\rangle\langle11|) + e^{-i\lambda}\left(|01\rangle\langle01| + |10\rangle\langle10|\right)\right] H \otimes H. \tag{B8}$$

The term in brakets corresponds to the spectral decomposition of $\exp\left[i\lambda(Z \otimes Z)\right]$, so that

$$\exp\left[i\lambda(X \otimes X)\right] = H \otimes H\left(C_X^{0\to1}\right)\mathbb{I} \otimes R_Z(-2\lambda)\left(C_X^{0\to1}\right)H \otimes H, \tag{B9}$$

and analogously for the other term

$$\exp\left[i\lambda(Y \otimes Y)\right] = S \otimes S\exp\left[i\lambda(X \otimes X)\right]S^\dagger \otimes S^\dagger. \tag{B10}$$

Figure 16 (b) illustrates the quantum circuit corresponding to $e^{i\lambda(X\otimes X)}$, while Fig. 16 (c) depicts the circuit for $e^{i\lambda(Y\otimes Y)}$.

## Appendix C: Formal analysis of Pessoa Júnior's experiment

As before, an initial state in which mode A is populated by one photon is considered, $|\psi_0\rangle = |1_A, 0_B, 0_E, 0_F\rangle$. After $BS_1$ the state is evolved to

$$|\psi_1\rangle = \frac{1}{\sqrt{2}}\left(|1_C, 0_E, 0_D, 0_F\rangle + i|0_C, 0_E, 1_D, 0_F\rangle\right); \tag{C1}$$

even though modes E and F are included for completeness, they remain non-populated. In the following, we present the most general states for the remaining optical elements. After $BS_4$ and $BS_5$ we have

$$|\psi_2\rangle = \frac{1}{2}\left(T_4|1_G, 0_H, 0_I, 0_J\rangle + iR_4|0_G, 1_H, 0_I, 0_J\rangle - R_5|0_G, 0_H, 1_I, 0_J\rangle + iT_5|0_G, 0_H, 0_I, 1_J\rangle\right), \tag{C2}$$

where the coefficients $T_4$ and $R_4$ ($T_5$ and $R_5$) correspond to the ability to adjust the transmission and reflection in $BS_4$ ($BS_5$) and $T_j^2 + R_j^2 = 1$ with $T_j, R_j \in \mathbb{R}$ for $j = 4, 5$. After $\phi_H$, the quantum state is

$$|\psi_3\rangle = \frac{1}{2}\left(T_4|1_G, 0_H, 0_I, 0_J\rangle + iR_4e^{i\phi_H}|0_G, 1_H, 0_I, 0_J\rangle - R_5|0_G, 0_H, 1_I, 0_J\rangle + iT_5|0_G, 0_H, 0_I, 1_J\rangle\right). \tag{C3}$$

After $BS_2$, we obtain the following state:

$$\begin{aligned}|\psi_4\rangle = \frac{1}{2}\left(T_4|1_G, 0_K, 0_L, 0_J\rangle + iT_5|0_G, 0_K, 0_L, 1_J\rangle\right) - \frac{1}{2\sqrt{2}}\left(R_5 + R_4e^{i\phi_H}\right)|0_G, 1_K, 0_L, 0_J\rangle \\ - \frac{i}{2\sqrt{2}}\left(R_5 - R_4e^{i\phi_H}\right)|0_G, 0_K, 1_L, 0_J\rangle.\end{aligned} \tag{C4}$$

The action of mirrors and $\phi_N = \pi$ leads to

$$\begin{aligned}|\psi_5\rangle = \frac{i}{2}\left(T_4|1_M, 0_N, 0_O, 0_P\rangle + iT_5|0_M, 0_N, 0_O, 1_P\rangle\right) + \frac{i}{2\sqrt{2}}\left(R_5 + R_4e^{i\phi_H}\right)|0_M, 1_N, 0_O, 0_P\rangle \\ + \frac{1}{2\sqrt{2}}\left(R_5 - R_4e^{i\phi_H}\right)|0_M, 0_N, 1_O, 0_P\rangle.\end{aligned} \tag{C5}$$

After $BS_3$, we get

$$\begin{aligned}|\psi_6\rangle = \frac{i}{2}&\left(T_4|1_M, 0_Q, 0_R, 0_P\rangle + iT_5|0_M, 0_Q, 0_R, 1_P\rangle\right) \\ &+ \frac{1}{4}\left[-\left(R_5 + R_4e^{i\phi_H}\right) + e^{i\phi_L}\left(R_5 - R_4e^{i\phi_H}\right)\right]|0_M, 1_Q, 0_R, 0_P\rangle \\ &- \frac{i}{4}\left[-\left(R_5 + R_4e^{i\phi_H}\right) - e^{i\phi_L}\left(R_5 - R_4e^{i\phi_H}\right)\right]|0_M, 0_Q, 1_R, 0_P\rangle.\end{aligned} \tag{C6}$$

## Appendix D: Step-by-step description of the modified Unruh's experiment with two-photons

### 1.  Development for the initial state $|11\rangle_{\mathrm{AB}}$

Consider the initial state $|11\rangle_{\mathrm{AB}}$. After the first BS, we have

$$|\psi_1\rangle = \frac{i}{\sqrt{2}} \left( |20\rangle_{\mathrm{CD}} + |02\rangle_{\mathrm{CD}} \right). \tag{D1}$$

After the mirrors and the phase $\phi_{\mathrm{E}}$ the state becomes

$$|\psi_2\rangle = -\frac{i}{\sqrt{2}} \left( e^{i\phi_{\mathrm{E}}} |20\rangle_{\mathrm{EF}} + |02\rangle_{\mathrm{EF}} \right). \tag{D2}$$

After BS$_2$, we get

$$|\psi_3\rangle = \frac{i\left(e^{i\phi_{\mathrm{E}}} - 1\right)}{2\sqrt{2}} \left( |20\rangle_{\mathrm{GH}} - |02\rangle_{\mathrm{GH}} \right) + \frac{1 + e^{i\phi_{\mathrm{E}}}}{2} |11\rangle_{\mathrm{GH}}. \tag{D3}$$

After the phase $\phi_{\mathrm{H}}$ and the mirrors, the state takes the form of

$$|\psi_4\rangle = \frac{1 - e^{i\phi_{\mathrm{E}}}}{2\sqrt{2}} \left( i\,|20\rangle_{\mathrm{IJ}} - ie^{i\phi_{\mathrm{H}}} |02\rangle_{\mathrm{IJ}} - \sqrt{2}e^{i\phi_{\mathrm{H}}} |11\rangle_{\mathrm{IJ}} \right) \tag{D4}$$

and finally, after BS$_3$, we get

$$|\psi_5\rangle = -\frac{i\sqrt{2}}{8} \left( 1 - e^{i\phi_{\mathrm{E}}} + 3e^{i\phi_{\mathrm{H}}} + e^{i\phi_{\mathrm{H}}}e^{i\phi_{\mathrm{E}}} \right) |20\rangle_{\mathrm{KL}} + \frac{i\sqrt{2}}{8} \left( 1 - e^{i\phi_{\mathrm{E}}} - e^{i\phi_{\mathrm{H}}} - 3e^{i\phi_{\mathrm{H}}}e^{i\phi_{\mathrm{E}}} \right) |02\rangle_{\mathrm{KL}}$$
$$- \frac{\left(1 - e^{i\phi_{\mathrm{E}}}\right)\left(1 - e^{i\phi_{\mathrm{H}}}\right)}{4} |11\rangle_{\mathrm{KL}}. \tag{D5}$$

### 2.  Blocking mode C

By blocking mode C, the normalized state that enters BS$_2$ is $|\psi_2'\rangle = -i\,|02\rangle_{\mathrm{EF}}$. So, after BS$_2$, we get

$$|\psi_3'\rangle = -i\left( \frac{1}{2} \left( |20\rangle_{\mathrm{GH}} - |02\rangle_{\mathrm{GH}} \right) + \frac{i}{\sqrt{2}} |11\rangle_{\mathrm{GH}} \right). \tag{D6}$$

After the phase $\phi_{\mathrm{H}}$ and mirrors

$$|\psi_4'\rangle = \frac{i}{2} |20\rangle_{\mathrm{IJ}} - \frac{ie^{i\phi_{\mathrm{H}}}}{2} |02\rangle_{\mathrm{IJ}} - \frac{e^{i\phi_{\mathrm{H}}}}{\sqrt{2}} |11\rangle_{\mathrm{IJ}} \tag{D7}$$

and finally, after BS$_3$,

$$|\psi_5'\rangle = -\frac{i}{4} \left\{ \left(1 + 3e^{i\phi_{\mathrm{H}}}\right) |20\rangle_{\mathrm{KL}} - \left(1 - e^{i\phi_{\mathrm{H}}}\right) \left[ |02\rangle_{\mathrm{KL}} + i\sqrt{2}\,|11\rangle_{\mathrm{KL}} \right] \right\}. \tag{D8}$$

### 3.  Blocking mode D

By blocking mode D, the normalized state that enters BS$_2$ is $|\psi_2''\rangle = -ie^{i\phi_{\mathrm{E}}} |20\rangle_{\mathrm{EF}}$. So, after BS$_2$, we get

$$|\psi_3''\rangle = \frac{ie^{i\phi_{\mathrm{E}}}}{2} |20\rangle_{\mathrm{GH}} - \frac{ie^{i\phi_{\mathrm{E}}}}{2} |02\rangle_{\mathrm{GH}} + \frac{e^{i\phi_{\mathrm{E}}}}{\sqrt{2}} |11\rangle_{\mathrm{GH}}. \tag{D9}$$

After the phase $\phi_{\mathrm{H}}$ and mirrors

$$|\psi_4''\rangle = -\frac{ie^{i\phi_{\mathrm{E}}}}{2} |20\rangle_{\mathrm{IJ}} + \frac{ie^{i\phi_{\mathrm{E}}}e^{i\phi_{\mathrm{H}}}}{2} |02\rangle_{\mathrm{IJ}} - \frac{e^{i\phi_{\mathrm{E}}}e^{i\phi_{\mathrm{H}}}}{\sqrt{2}} |11\rangle_{\mathrm{IJ}} \tag{D10}$$

and finally, after BS$_3$,

$$|\psi_5''\rangle = \frac{ie^{i\phi_\mathrm{E}}}{4} \left[ \left(1 - e^{i\phi_\mathrm{H}}\right) |20\rangle_\mathrm{KL} - \left(1 + 3e^{i\phi_\mathrm{H}}\right) |02\rangle_\mathrm{KL} - i\sqrt{2} \left(1 - e^{i\phi_\mathrm{H}}\right) |11\rangle_\mathrm{KL} \right]. \tag{D11}$$

---

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
