# Peer review of "Digital quantum simulation of bosonic systems and quantum complementarity"

_SciPost Physics Core, doi:SciPost Phys. Core 8, 086 (2025)_

## Round 1 · Referee Report · Anonymous (Referee 1) · 2025-5-14

Strengths

scipost_202503_00053v1

The Afshar experiment, a variation of the double-slit experiment in quantum mechanics, was devised and conducted by Shahriar Afshar in 2004. Afshar asserted that the experiment provides information about the path a photon takes through the apparatus while simultaneously allowing for the observation of interference between the paths. He claimed that this finding contradicts the complementarity principle of quantum mechanics. In response, several experiments have critiqued Afshar's work, including one by Unruh. Unruh, like Kastner, set up a configuration he believes is both equivalent and simpler. His approach magnifies the effect, making it easier to identify flaws in the logic. This article presents a detailed overview of Afshar's original experiment, along with Unruh's experiment and a modified version (Pessoa Junior's), to examine the issues raised by Afshar concerning a potential violation of the complementarity principle.

The authors have digitally simulated interferometric variants of Afshar's experiment using IBM's quantum computers. They have explored the analogous experiments conducted by Unruh and Pessoa Júnior, delving into discussions regarding the apparent violation of Bohr's complementarity principle in the context of the entire experimental setup. Additionally, they have analyzed these experiments within the framework of an updated quantum complementarity principle, which applies to specific quantum state preparations while remaining consistent with the foundational principles of quantum mechanics.

I believe that much of the calculations, except for minor errors (e.g. A1 to A5) and analysis is fine. However, I think using a digital simulator on an IBM machine may be a little heavy handed. The analysis could proceed with simple optical experiments on an optical bench, although it may somehow lose some oomphs and also some of the experiments may have already been demonstrated. So, aside from the usual arguments from the proponents and critiques of the original exeriment, what really is new aside from a somewhat trivial application of a quantum simulator.

Overall, I think the paper is more suitable for a pedagogical journal. I do not recomment it for SciPost.

Report

I think the article is better suited for a pedagogical journal.

Recommendation

Accept in alternative Journal (see Report)

  • validity: ok
  • significance: ok
  • originality: low
  • clarity: good
  • formatting: acceptable
  • grammar: acceptable

Author:  Diego Samuel Starke  on 2025-05-19  [id 5497]

(in reply to Report 1 on 2025-05-14)
Category:
answer to question
reply to objection

We appreciate the Referee's willingness to review our manuscript. But we respectfully disagree with Referee #1 overall evaluation for the following reasons. We have organized the Referee's remarks into the three outlined items for clarity:

  1. I believe that much of the calculations, except for minor errors (e.g. A1 to A5) and analysis is fine;
  2. However, I think using a digital simulator on an IBM machine may be a little heavy handed. The analysis could proceed with simple optical experiments on an optical bench, although it may somehow lose some oomphs and also some of the experiments may have already been demonstrated;
  3. So, aside from the usual arguments from the proponents and critiques of the original exeriment, what really is new aside from a somewhat trivial application of a quantum simulator. Overall, I think the paper is more suitable for a pedagogical journal. I do not recomment it for SciPost.

Regarding item 1: We respectfully ask the referee to specify the specific errors found in Eqs. A1 to A5. Despite conducting a comprehensive review ourselves, we were unable to detect any mistakes in these equations.

Regarding item 2: Although we make it clear in the manuscript that the use of the formalism of Mohan et al. is essential only for Pessoa Júnior's setup, it is important to point out that not even Unruh's experiment has been carried out using optical devices to date, at least to our knowledge. The reason behind this is precisely the experimental difficulty in an optical bench of even running the usual Mach-Zehnder interferometer (MZI), which are preferably performed in analog versions such as the Sagnac interferometer. Although the modified version of the Unruh experiment is feasible using optical instruments, such as the experiment performed in Ref.~[1], Pessoa Júnior's experiment is much more challenging, and, as far as we know, there are no experiments that resemble this setup. It is also worth mentioning that digital simulations can be extended to even more complicated experimental setups, with a larger quantity of optical devices and higher occupation numbers, whose implementations (Ref.~[2]) are difficult and relatively novel in the field.

Regarding item 3: Our article is completely built around three points: $(i)$ extending and employing the framework established by Mohan et al. to $(ii)$ conduct a thorough examination of two comparable versions of Afshar's experiment: a modified version of Unruh's experiment and Pessoa Júnior's experiment, and $(iii)$ examining these experiments with regard to the newly proposed quantum complementarity principle. All of these points are new in the literature. Our research is based on the recent methodologies outlined by Mohan et al. . So, we are confident that our work has the relevance and novelty needed for justify its publication in SciPost.

Attachment:

Reply_to_Ref.1.pdf

---

## Round 1 · Referee Report · Anonymous (Referee 2) · 2025-6-16

Report

In this manuscript, the authors address the digital simulation of quantum systems based on bosonic degrees of freedom by means of state-of-the-art quantum computers. In particular, they discuss the quantum complementary principle in the context of two experiments proposed by Pessoa Júnior. These experiments aim to both simulate the main features of Afshar’s experiment and provide an interpretation in the framework of the complementary principle. Therefore, the authors provide a comparison between theoretical probabilities, classical simulations and demonstrative results obtained by means of IBMQ quantum computers. The manuscript presents broad reviews of the mathematical background, yet it lacks in the exposition of valid new results. To motivate this statement, I list three main reasons below:

  1. The authors have extensively illustrated the technique introduced by Mohan et al., extending the method to a set-up embracing more than two bosons. In Mohan et al., the bosonic nature of the particles they simulated covers a fundamental role, since they were interested in the simulation of the HOM effect. On the contrary, all experimental schemes discussed in this manuscript require only one quantum excitation in total. The high (infinite) dimensionality of the Hilbert space of bosonic degrees of freedom hence does not seem to play any relevant role in the discussion, since the presence (or absence) of the quanton in a specific mode can all in all be described in terms of single qubit modes without invoking any bosonic framework. The authors should better motivate the choice of extending and employing such mathematical apparatus.

  2. In both Fig.8 and Fig.11 demonstrations and theoretical/simulation data show important fluctuations. Authors claim that this is due to the presence of noise and losses. Nevertheless, these discrepancy sources have an impactful effect on the outcomes. In some histogram the discrepancy between the expected and the observed data even amounts to over 40%, such as in Fig.11A in 0001, where the expected value is 0.02, whereas the demonstration displays the value 0.011. In Fig.8 the demonstrations display fluctuations as wide as or even wider than the expectation values in other plots. These results seem to lead to the conclusion that the instruments used for the demonstrations are not enough accurate for this kind of measurements.

  3. A last relevant remark concerns their analysis in the light of the quantum complementarity principle. In their discussions, authors recall and report the results achieved in [37], where the quantum complementarity principle has been applied to both the BMZI and the Unruh experiment. Moreover, they claim that “The explanation of the experiment in Fig. 2 using the QCP also extends to the modified Unruh’s experiment (Fig. 4) and Pessoa Júnior’s experiment (Fig. 9). This follows from the fact that BBS1 in these setups is a generalization of BS1, which corresponds to the specific case where T1 = R1 = 1/√2. Since all these experimental configurations are designed to analyze the quanton’s dual behavior after the first beam splitter, their description follows naturally from the QCP framework”. This seems to be the core of their theoretical results. Yet, they do not provide any further concrete mathematical analysis to support it. Moreover, the BMZI, which includes biased BS, has already been discussed in [37]. If both the description of both the modified Unruh’s experiment and Pessoa Júnior’s experiment follows from the QCP framework so naturally that it does not even require a more detailed analysis, it is not clear what are the elements of novelty of the manuscript.

Therefore, I would not recommend the publication unless the authors properly address the issues presented above.

Finally, some relatively minor issues related to the readability of the manuscript: 1) In Fig.3 detector D(A) and D(B) should be switched. Or the sentence on page 5 and the caption of Fig.3 must be changed. According to both Afshar’s experiment and the current notation on the figure, closing the slit A we should observe a click on the detector B. 2) What phase phi_H and phi_E have been used for the histograms in fig.8? Are they phi_E=1/2 and phi_H=0? 3) In the end of page 12, state B_1 has six digit. 4) In Fig.10, the action of BS_2, U_M and BS_3 seems to involve all four modes, whereas it actually involves only two of them. For this reason, as an attachment I prepared an alternative scheme I believe more suitable for fig.10. The authors may consider this scheme instead of that in Fig.10. If not, they should better explain the switch between P and M after U_M.

Attachment

Recommendation

Ask for major revision

---

## Round 2 · Referee Report · Anonymous (Referee 1) · 2025-10-5

Disclosure of Generative AI use

The referee discloses that the following generative AI tools have been used in the preparation of this report:

Not applicable

Strengths

I believe that the authors have addressed some of my concerns. I still believe that the authors could elaborate some parts substantially and submit it as a pedagogical review.

Weaknesses

However, as mentioned in my earlier report, I do not think that the content of the manuscript is sufficiently novel. I note the difficulty with optical set-ups but then I am not entirely convinced that a proof using a noisy quantum computer suffices to address all the concerns of Afshar experiment. Also, I am not sure if the authors have "addressed an important (set of) problem(s) in the field using appropriate methods with originality" nor "detail one or more new research results significantly advancing current knowledge and understanding of the field". The only novelty is the implementation on an IBM quantum computer.

Report

I would still recommend that the article be substantially elaborated with the history of the arguments and the current contributions for a comprehensive narrative of the Afshar experiments and its subsequent controversy and improvements.

Requested changes

Not applicable. (Yes, A1-A5 seems Ok)

Recommendation

Ask for major revision

---

## Round 2 · Referee Report · Anonymous (Referee 2) · 2025-10-6

Strengths

Authors addressed all points raised in my previous report and carried out important changed in the manuscript. Importantly, they both utilized a different instrument (drastically improving their data set), and exploited the potential of their model by considering two bosons.

Report

The new version fulfills the Journal's acceptance criteria.

Recommendation

Publish (meets expectations and criteria for this Journal)

---

## Round 2 · Author Response

Dear Editor, thank you for your e-mail regarding our manuscript entitled ``Digital quantum simulation of bosonic systems and quantum complementarity''. We would also like to thank you for the opportunity to do a major revision. Following Referee #2's comments, we realized that our manuscript could perhaps be improved, as it was, and that it consequently reflects the points raised by Referee #1. With the contributions of both Referees, we believe our work has been improved. Therefore, we thank both Referees for the points raised. Please find below our response to the Referees' comments. Finally, a list of the changes made in the manuscript is included.

Our response to Referee #1: In addition to what we have already answered, it follows that: The above considerations are preliminary responses before Referee #2's review is released. Referee #2's opinion touches on the points raised by Referee #1, but in a more comprehensive manner. Based on both opinions, we introduce additional elements that we believe address Referee #1's questions in the same way. Therefore, we invite Referee #1 to follow the response to Referee #2.

Our response to Referee #2: We appreciate the Referee's willingness to carefully review our manuscript. We consider the review to be comprehensive and to have prompted a number of inquiries that require further elucidation. These inquiries, we are confident, will certainly contribute to enhancing and refining our manuscript. It is our intention to address the inquiries raised as they are of significant relevance.

Regarding item 1: We agreed with the Referee that we are not using more than one quanton which goes in a different line from the intention of the formalism used by Mohan et al. to simulate the HOM effect. Although that formalism was developed with a different purpose in mind, our extension builds upon it to enable the analysis and treatment of more intricate and sophisticated optical setups, such as those employed in the experiment by Pessoa Júnior. Even though our implementation does not involve multiple quantons, the extension of the formalism we develop remains equally suitable for scenarios where they are present. Additionally, to our knowledge, implementing the simulations is feasible for the standard versions of the interferometers, but becomes challenging with the addition of blockers. As the formalism separates the problem into modes, with single photons the same state that represents the arm where the blocker is inserted is packed in the qubit. An initial approach discussed in this manuscript involves focusing on the single qubit until it enters the blocker region (discarding other components), disregarding statistics for cases not passing through the blocker, and recording the data. In a subsequent experiment, we must manually set the post-blocker state and proceed with the experiment as if the ``photon'' traversed the blocker region. We will then collect data from before and after the blocker to present the experimental conditions. It seems feasible to us with the aid of an auxiliary qubit for correlation, but we failed to find a better version than the one presented in the manuscript. Therefore, it was precisely this scenario that directed us to Mohan et al.'s and other digital simulation papers and ultimately to the present manuscript. Thus, with just a single circuit, we can somewhat simulate the experiment as a whole without needing to make the kind of considerations we mentioned earlier. Personally, we find this approach less artificial than the previous one, which requires two distinct configurations and the input of a manufactured state. In our approach, we simply record the measurement in the blocker and reset the qubit's mode to show it has been ``absorbed'' by the blocker. However, even in cases where it is not ``absorbed'', the experiment remains valid and captures the scenario where the photon is not ``absorbed'', which seems to be much more similar to what happens in a bench experiment.
We include a few sentences along the text to emphasize this feature of the experiment. Despite that and building on your questioning, we introduce an initial state in the modified Unruh's experiment that occupies both modes and follow with a discussion similar to that presented by Pessoa Júnior. Therefore we include section E ``Modified Unruh's experiment for the two-photon case'', which is truly innovative in this regard and simultaneously addresses Referee \#1's questions.

Regarding item 2: At the time the demonstrations on quantum hardware were conducted, we only had access to IBM's open ``Eagle'' processors, which are not the best in terms of noise and gate error. However, we recently earned credits from that company to be spent on any of the available hardware, including the ``Heron'' type, which so far has proved to be substantially better than earlier processors in the demonstrations we have conducted for other works. With that in mind, we ran some demonstrations on better hardware and observed a significant improvement in the results, and because of that, we carried out the demonstrative results again. Error bars have been included in the new version. We can see that the results have been significantly improved with the use of higher-quality hardware.

Regarding item 3: We agreed with the Referee that some elements are missing to enrich the section. A significant missing component in the theoretical discourse is the omission of the critique on retro-inference methods utilized by Pessoa Júnior. Although this is mentioned within the manuscript, it is not revisited in this section to develop a more comprehensive critique. The QCP framework brings an updated view of this type of experimental setup, which is, in some aspects, new in the literature. Pessoa Júnior extrapolates the most well-known phase variation to infer the wave behavior (also particle behavior) in the quantitative Bohr's complementarity principle. In addition to the QCP framework being similarly utilized in this scenario, the analysis is innovative and sheds light on a fundamental discussion within quantum mechanics, but was clearly not well employed, as noted by the Referee. To fill this gap, we discuss a new version of the delayed-choice experiment that arises when we apply the extrapolation methods used by Pessoa Júnior. This setup, besides being a new experiment, increases the discussion on retro-inference and further emphasizes the lack of a formal definition of Bohr's complementarity principle. Additionally, we explore the QCP analysis for the two-photon case. This analysis also goes beyond what we have previously discussed on the subject. Therefore, we believe this result also greatly enriches the manuscript and closes the gap highlighted by the Referee in this section. It is worth noting that we performed quantum state tomography by reconstructing $\rho$ using the mean values of the Pauli basis observables. Even using classical simulation, our results were not as good for predictability as they were for coherence; the results can be seen in Fig. 15. However, the results in the classical simulation were only relatively good when we used $2^{23}$ shots in $100$ Trotter steps. Unfortunately, it is completely unfeasible to perform this type of experiment on current quantum hardware. As we saw, in our simplest case, we already had problems obtaining the results.

Regarding minor issues: 1) The sentence in the manuscript contained an error and has been corrected to display as $D_B$ ($D_A$) where it was previously written as $D_A$ ($D_B$). 2) In the graphs in Fig.~7 we have used $\phi_{\text{E}} = \pi/2$ and $\phi_{\text{H}} = 0$ and we have included a sentence in the subsequent text to make everything more consistent. 3) Thank you for noticing this mistake too, the right version is $\left\vert {B_{1}}\right\rangle =-\frac{R}{2}\left\vert 01000\right\rangle_{\text{DLMQR}}$ and it was also corrected. 4) In fact, we uploaded an unintended image and we have corrected this issue as well. Thank you for the circuit version. Although we do not use it, we will try to explain this point better based on the updated version. The intended version has some markings on the unitary block, labeled $``0"$ and $``1"$, to highlight which qubits are involved in the operation and is the usual form used by Qiskit.

---

## Round 2 · List of Changes

• The abstract has undergone changes to reflect the major revision;
  • We changed the final three paragraphs of the Introduction to emphasize the content and incorporate the changes following this Reply;
  • All paragraphs on page 3 have been modified to bring the content more in line with the use of the two-photon case or more bodies and to assist in future implementations that may be based on this work;
  • The last paragraph on page 7 has been changed;
  • Because of the modifications on page 3, equations (26)-(29) have been adjusted accordingly;
  • The first paragraph on page 9 has undergone changes;
  • We added two more sentences to the end of figures 7, 8, and 11;
  • The penultimate chapter on page 11 has undergone a major revision;
  • Every paragraph on page 13 has undergone a major revision;
  • We included Section E that comprehends the modified Unruh's experiment for the two-photon case;
  • The Figs.~12 and 13 arise in the discussion of Section E;
  • Section F is modified to include the Subsections 1 (about delayed choice experiments) and 2 (to analyze the two-photon case).
  • Due to the expansion of Section F, Figs.~14 and 15 are inserted due to the new version of the delayed choice experiment and also due to the classical simulation of the QCP for the two-photon case;
  • The Appendix D was changed. The new Appendix D is included to present the state evolution for the two-photon case. The supplemental material [63] now includes the previous version of the Appendix D, presenting the manuscript's results with error bars.
  • We made some minor edits throughout the manuscript to correct Referee nominations and other minor details.

---

## Editorial Decision

published